# Buffered Qualitative Stability explains the robustness and evolvability of transcriptional networks

Luca Albergante[1]*, J Julian Blow[1]*[†], Timothy J Newman[1,2]*[†]

[1]College of Life Sciences, University of Dundee, Dundee, United Kingdom; [2]School of Engineering, Physics and Mathematics, University of Dundee, Dundee, United Kingdom

**Abstract** The gene regulatory network (GRN) is the central decision-making module of the cell. We have developed a theory called Buffered Qualitative Stability (BQS) based on the hypothesis that GRNs are organised so that they remain robust in the face of unpredictable environmental and evolutionary changes. BQS makes strong and diverse predictions about the network features that allow stable responses under arbitrary perturbations, including the random addition of new connections. We show that the GRNs of *E. coli*, *M. tuberculosis*, *P. aeruginosa*, yeast, mouse, and human all verify the predictions of BQS. BQS explains many of the small- and large-scale properties of GRNs, provides conditions for evolvable robustness, and highlights general features of transcriptional response. BQS is severely compromised in a human cancer cell line, suggesting that loss of BQS might underlie the phenotypic plasticity of cancer cells, and highlighting a possible sequence of GRN alterations concomitant with cancer initiation.

**\*For correspondence:**
l.albergante@dundee.ac.uk (LA);
j.j.blow@dundee.ac.uk (JJB);
t.newman@dundee.ac.uk (TJN)

[†]These authors contributed equally to this work

**Competing interests:** The authors declare that no competing interests exist.

**Reviewing editor**: Detlef Weigel, Max Planck Institute for Developmental Biology, Germany

## Introduction

At every level of organisation, biological entities, such as genes, proteins and cells, function as ensembles. Interaction networks are therefore a fundamental feature of biological systems, and a vast amount of analysis exploring the organisation of biological networks has been performed (*Milo et al., 2002*; *Barabasi and Oltvai, 2004*; *Alon, 2006*; *Buchanan et al., 2010*). This analysis has provided interesting insights into the features of these networks (*Barabasi and Oltvai, 2004*; *Brock et al., 2009*; *Tyson and Novák, 2010*; *Ferrell et al., 2011*; *Liu et al., 2011*; *Cowan et al., 2012*), and has led to new methodologies for characterizing their topologies. However, one might argue that this work has had less impact on our understanding of the reasons underlying the network topologies observed and on the possible selective pressures leading to the emergence of common network features. Here we present a simple theory, Buffered Qualitative Stability (BQS), motivated by biological robustness, which has strong explanatory power and provides a number of hard, readily verifiable predictions for the topological structure of interaction networks, at both global and local scales. Besides leading to new predictions—that are consistently verified—BQS provides a theoretical justification for the ubiquitousness of network features already observed. BQS is therefore an important step in providing a general mechanistic explanation for the overall structure of GRNs at different scales and in shedding new light on previous observations.

Robustness is a remarkable feature of living organisms allowing them to tolerate a wide variety of contingencies, such as DNA damage, limitations in nutrient availability, or exposure to toxins (*Lopez-Maury et al., 2008*; *MacNeil and Walhout, 2011*). Although much is now known about how cells respond to particular stresses or environmental cues, little is known about how cells remain stable and respond appropriately whatever the contingency. Over evolutionary time it is also advantageous for organisms to be robust to genetic changes, including those that occur as a consequence of the shuffling of genes during sexual reproduction. In order for cells to be fully robust, changes to any of the thousands

**eLife digest** The genomes of living organisms consist of thousands of genes, which produce proteins that perform many essential functions. Cells receive signals from both their internal and external environments, and respond by changing how they express their genes. This allows a cell to make the right amount of different proteins when needed. The proteins that a cell produces can then, in turn, influence how the cell's genes are expressed. This set of interactions between genes and proteins is called a gene regulatory network, and is akin to a computer program that the cell runs to define its behaviour. At present, we understand very little about why these networks take on the forms seen in living cells.

A remarkable feature of living organisms is their ability to withstand an extremely wide variety of predicaments, such as DNA damage, physical trauma or exposure to toxins. This ability, generally called robustness, requires a cell to rapidly activate different gene sets and maintain their activity for as long as necessary. However, very little is known about how cells are programmed to respond appropriately, whatever happens, and keep themselves in a stable state.

Albergante et al. propose that a fully robust gene regulatory network should be able to stabilize itself. This means that the robustness of a gene regulatory network should only depend on how it is wired up, and not on quantitative changes to any features that may change unpredictably—for example the concentration of a protein.

By analysing data that is already available about gene regulatory networks in a wide selection of organisms ranging from bacteria to humans, Albergante et al. show that all known gene regulatory networks are wired up in a way that any quantitative change to the network will not cause the state of network to change. In addition, gene regulatory networks tend to remain stable even if new regulatory links are randomly added. Albergante et al. call this property Buffered Qualitative Stability (BQS): the network is *qualitatively stable* because its state does not change when the activity of particular regulatory links in the network changes, and it is *buffered* against its stability being compromised by the random addition of new links.

Albergante et al. also found that the gene regulatory network of a cancer cell does not match up with the predictions of BQS, suggesting that the robustness of the network is compromised in these cells. This could explain why cancer cells are able to easily change their characteristics in response to changes in the environment. In addition, using BQS to analyse the gene regulatory network of bacteria such as *E. coli* reveals points in the network that, if disrupted, would make the network unstable, potentially harming the cell. Therefore, in the future, an understanding of BQS could help efforts to design new drugs to treat a range of infections and diseases.

of individual quantitative parameters—for example the concentration of a transcription factor or its affinity for its cognate DNA sequence—cannot be critical because contingencies may cause these to change. We propose that the robustness of a biological system should therefore depend on qualitative, not quantitative, features of its response to perturbation.

Robustness is a complex and fundamental feature that can be formalised in many ways (*Jen, 2003*; *Silva-Rocha and de Lorenzo, 2010*). Features commonly associated with robustness include resistance to noise, redundancy and error-correction. Here we will focus on an important component of robustness: the ability of a system at equilibrium to respond to a perturbation by returning to its equilibrium state. Such a feature, generally called 'stability', is essential to allow a system to properly operate in noisy conditions and withstand unexpected environmental challenges.

This type of robustness has been studied before (*Quirk and Ruppert, 1965*; *Puccia and Levins, 1985*) and has been applied to economics (*Quirk and Ruppert, 1965*; *Hale et al., 1999*), ecology (*May, 1973a*; *Puccia and Levins, 1985*) and chemistry (*Tyson, 1975*). However, this notion has never been used to predict network features beyond simple topological properties required by the 'rules' that allow such stringent robustness and has not previously been applied to molecular cell biology, or the evolutionary pressures shaping the behaviours of living organisms.

Transcriptional regulation plays a central role in the behaviour of cells in response to environmental cues and is aberrant in many diseases (*Lee and Young, 2013*). Moreover, networks representing

transcriptional interactions have been derived for diverse organisms. These networks, termed 'gene regulatory networks' (GRNs), comprise directed links between pairs of genes. For a given pair of linked genes, one encodes a transcription factor (TF) that regulates the expression of the other (*Buchanan et al., 2010*). *Figure 1* shows the GRN of *Escherichia coli*, with TFs coloured red and the arrows colour-coded according to the number of genes regulated by the source TF. Systematic network analysis of GRNs is possible because comprehensive and high quality GRN datasets are available for different organisms (*Lee et al., 2002*; *Harbison et al., 2004*; *Luscombe et al., 2004*; *MacIsaac et al., 2006*; *Galan-Vasquez et al., 2011*; *Sanz et al., 2011*; *Garber et al., 2012*; *Gerstein et al., 2012*; *Salgado et al., 2012*).

Here we consider the hypothesis that to confer robustness and promote evolvability, GRNs must be stable to changes in interaction parameters and also stable to the addition of new regulatory links, that is to changes in the structure of the GRN itself. The type of robustness that we hypothesize ensures that the transcriptional state of a cell remains largely stable in response to random perturbations. We show that published GRNs, including those of *E. coli*, *M. tuberculosis*, *P. aeruginosa*, *S. cerevisiae*, mouse and humans are robust in this way, a property we term 'Buffered Qualitative Stability' (BQS). Remarkably, the only published GRN of a cancer cell line deviates strongly from BQS, suggesting that the loss of BQS may play an important role in cancer.

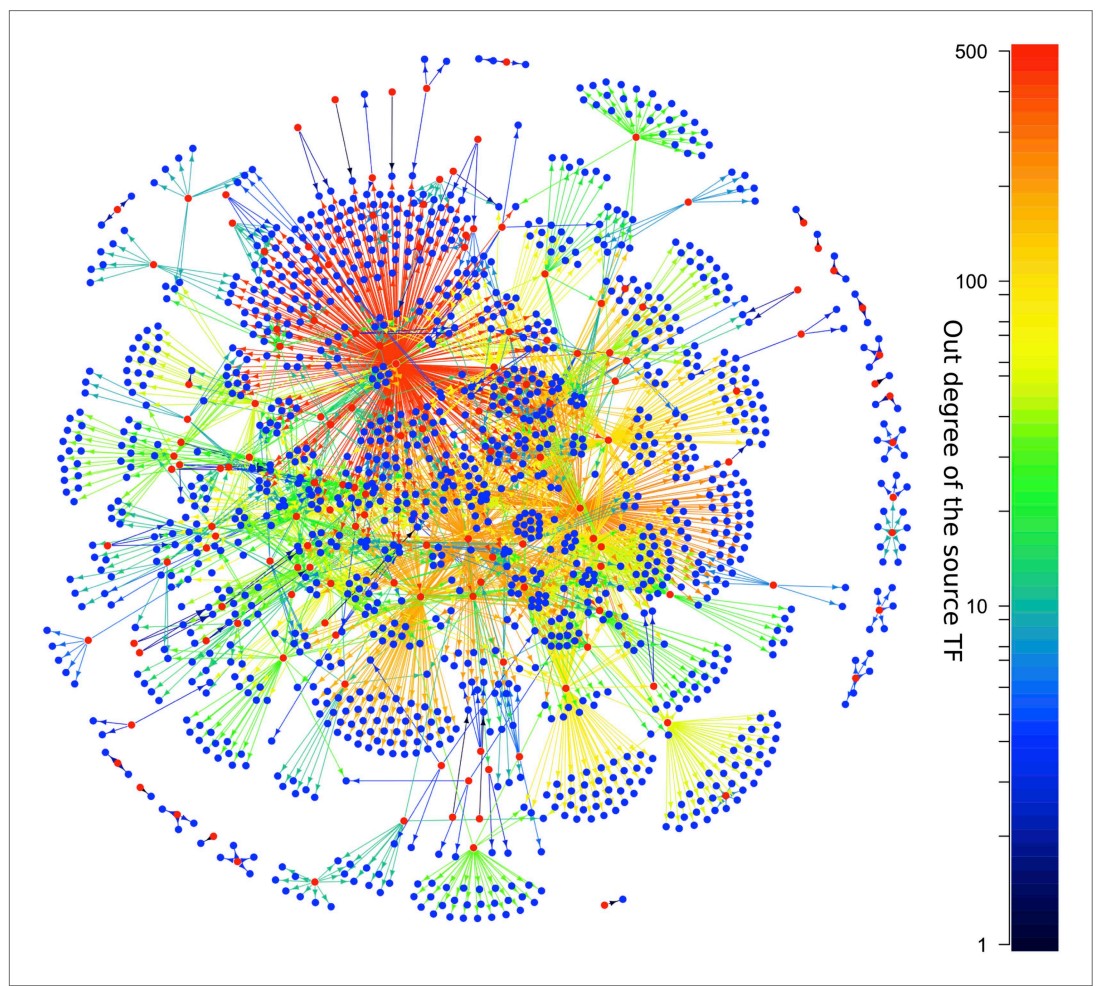

**Figure 1**. The *E. coli* GRN. The *E. coli* GRN derived from *Salgado et al. (2012)* using two evidence codes. Genes that are reported to regulate transcriptionally at least one other gene, that is transcription factors (TFs), are represented as red circles; the other genes are represented by blue circles. Arrows indicate a transcriptional interaction from the TF to the target gene. The arrows are colour-coded according to the number of genes regulated by the source TF. Note the logarithmic scale in the colour coding.

## Results

### GRNs are qualitatively stable

Interaction networks are ubiquitous in biology, and robustness in their response to perturbation is a desirable property in many circumstances. A mathematical theory called 'Qualitative Stability' has determined how the topological structure of a network is related to robustness (*Quirk and Ruppert, 1965*). This theory, discussed in 'Materials and methods', shows that certain network topologies remain stable even if the strength of any of the network interactions is varied in an arbitrary way. Qualitatively Stable GRNs would be robust, for example, to changes in the concentration of a transcription factor or its affinity for its cognate DNA sequence.

A primary requirement for Qualitative Stability is the absence of long feedback loops (meaning, in the case of GRNs, feedback loops involving three or more genes) regardless of whether the connections comprising the loops are stimulatory or inhibitory. In addition, 2-node feedback loops can be Qualitatively Stable depending on the precise nature of their interactions (see 'Materials and methods'). The danger inherent in feedback loops was first analysed by James Clerk Maxwell who showed that mechanical governors regulating the output of steam engines can fail if the input changes faster than the system response, causing 'an oscillating and jerking motion, increasing in violence till it reaches the limit of action of the governor' (*Maxwell, 1868*). Because there is an inevitable time lag between a transcription factor (TF) binding to the promoter of a gene and the production of the protein product of that gene, this form of instability can occur if GRNs contain feedback loops consisting of three or more TFs. This concept is supported by the behaviour of the repressilator, a well-known gene circuit consisting of a 3-gene feedback loop, which has been shown to produce oscillations of increasing intensity in vivo (*Elowitz and Leibler, 2000*).

We therefore examined the structure of organisms for which system-wide GRNs have been published, including three bacteria—*E. coli* (*Salgado et al., 2012*), *M. tuberculosis* (*Sanz et al., 2011*), *P. aeruginosa* (*Galan-Vasquez et al., 2011*)—the yeast *S. cerevisiae* (*Harbison et al., 2004*), and human (represented by the GM12878 cell line) (*Gerstein et al., 2012*). The main Figures present data from *E. coli*, *S. cerevisiae* and human, whilst analysis of *M. tuberculosis* and *P. aeruginosa* plus additional confidence levels for *E. coli* and *S. cerevisiae*, and additional yeast datasets (*Lee et al., 2002*; *Luscombe et al., 2004*; *MacIsaac et al., 2006*) are reported in the Figure Supplements and confirm our main results. A review of available datasets and the rationale for our selection is given in 'Materials and methods'.

On studying feedback loops in the GRNs of these organisms (*Figure 2A–C*, *Figure 2—figure supplement 1A,B*, lightly shaded bars), we find that *P. aeurginosa*, *S. cerevisiae* and human GRNs have no feedback loops comprising three or more genes. The *E. coli* GRN has no feedback loops comprising four or more genes, and only two 3-gene feedback loops. *M. tuberculosis* has two 3-gene feedback loops and one 4-gene feedback loop. Notably, all the 3-gene feedback loops observed in real GRNs share the same peculiar structure, with implications discussed below. In contrast, when networks of the size and connectivity of the biological GRNs are constructed with randomly placed links, they display an exponential increase in feedback loops consisting of three or more genes, which number in the thousands (*Figure 2A–C*, *Figure 2—figure supplement 1A,B*, heavily shaded bars, and *Figure 2—figure supplement 4B*). Each of 1000 randomly simulated *E. coli* networks had at least one long feedback loop. The vastly different abundances of feedback loops clearly demonstrate the profound difference in topologies between real and random networks. Statistical analyses suggest that there is an extremely small probability ($<10^{-6}$) that the absence of long feedback loops with >3 genes in *E. coli* is a chance event (*Figure 2—figure supplement 2A*). Similar results hold for *S. cerevisiae* (*Figure 2—figure supplement 2B*) and human (*Figure 2—figure supplement 2C*). These results are robust to variations in the confidence levels of the *E. coli* and *S. cerevisiae* GRNs (*Figure 2—figure supplement 3E,J*), despite large variations in other properties of the GRNs (*Figure 2—figure supplement 3A–D,F–I*), and remain valid when different random models are considered (*Figure 2—figure supplement 4B*).

Qualitative Stability of GRNs prevents the catastrophe described by Maxwell from occurring, even when any of the myriad quantitative system parameters (TF abundance, promoter availability, the rate of transcription, etc.) are altered. This stability provides a type of 'damping', which will tend to restore the state of the GRN when challenged with contingencies that might otherwise induce chaotic or unpredictable behaviour. Notably, the damped oscillatory response expected from a stable system has been observed in single cell experiments (*Tay et al., 2010*).

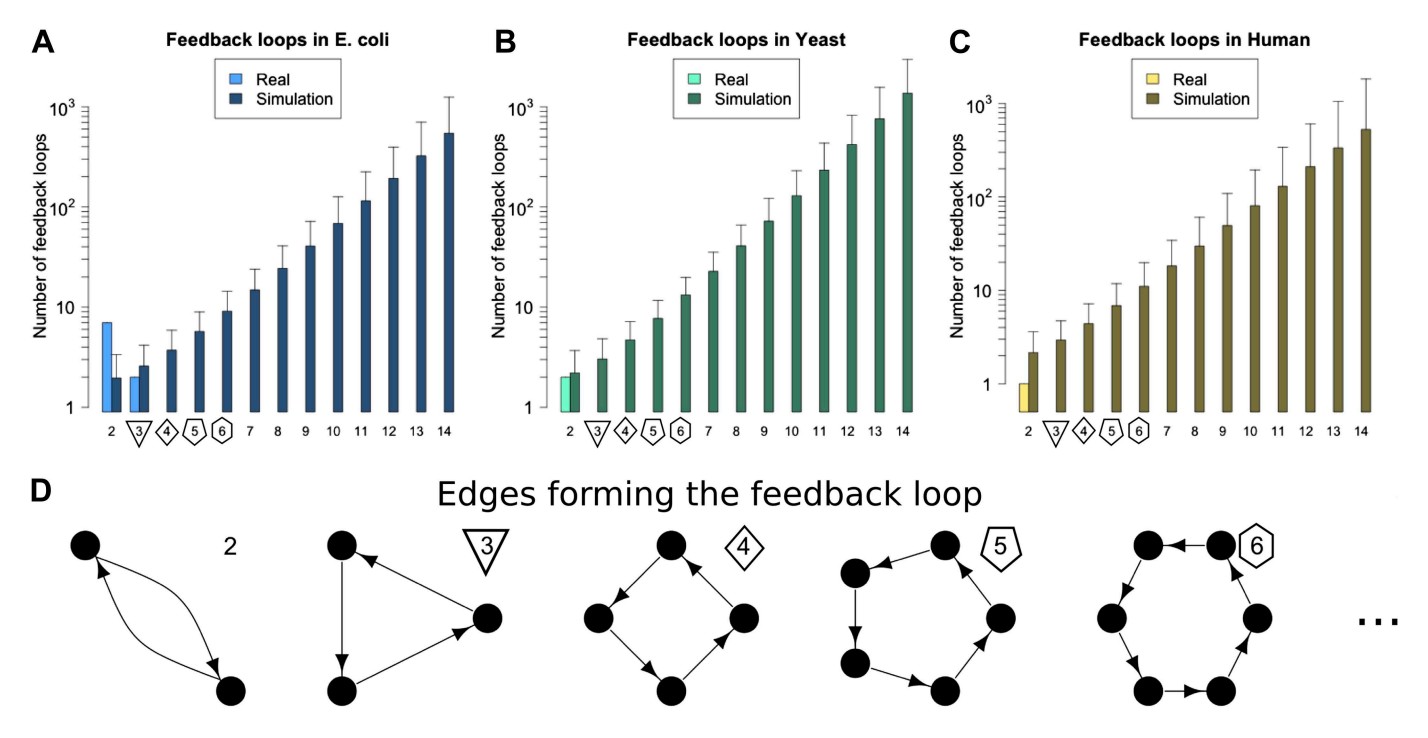

**Figure 2**. Feedback loops in real and simulated GRNs. Number of feedback loops is provided on a logarithmic scale for *E. coli* (**A**), *S. cerevisiae* (**B**), and the human GM12878 cell line (**C**) and in each case is compared with a randomly simulated network containing the same number of genes, TFs, and connections. For the random networks, each graph reports the mean and standard deviation. (**D**) Schematic illustrations of feedback loops of length 2–6 are plotted in black.

The following figure supplements are available for figure 2:

**Figure supplement 1**. Feedback loops in *M. tuberculosis*, *P. aeruginosa* and other yeast datasets.

**Figure supplement 2**. p-Value estimation for the number of long feedback loops.

**Figure supplement 3**. General properties and number of feedback loops in the RegulonDB (*E. coli*) and Harbison et al. (yeast) datasets under different statistical conditions.

**Figure supplement 4**. General properties and number of feedback loops under different random model for RegulonDB (*E. coli*).

## BQS predicts large-scale properties of GRNs

In principle the Qualitative Stability observed in GRNs might be easy to break by addition of another link to the network. For example, a long feedback loop can be created by the addition of a feedback connection from a TF lower down in a network path to a TF higher up in that path. This could occur, for example, through a mutation in the promoter of the target gene allowing it to be bound by a new TF. It is also possible that stress conditions could cause TFs to act inappropriately at promoters they do not normally regulate. In this way Qualitative Stability could be lost, and GRNs could become unstable. Thus, we predict that if long feedback loops are detrimental because of their instability, then GRNs would be configured to minimize destabilization via the addition of new connections. We call network paths that can be transformed into loops by the addition of a single new link 'incomplete feedback loops' (*Figure 3D–E*). The abundance of incomplete feedback loops in the GRNs of *E. coli*, *S. cerevisiae* and human is shown in *Figure 3A–C* (lightly shaded bars). Data for *M. tuberculosis* and *P. aeruginosa* are shown in *Figure 3—figure supplement 1A,B*. For each of these GRNs there are <2000 incomplete feedback loops and they tend to be of a relatively small size. A similar empirical observation has been made regarding transcriptional cascades (*Rosenfeld and Alon, 2003*; *The modENCODE Consortium et al., 2010*). This is in stark contrast to comparable random networks (*Figure 3A–C*,

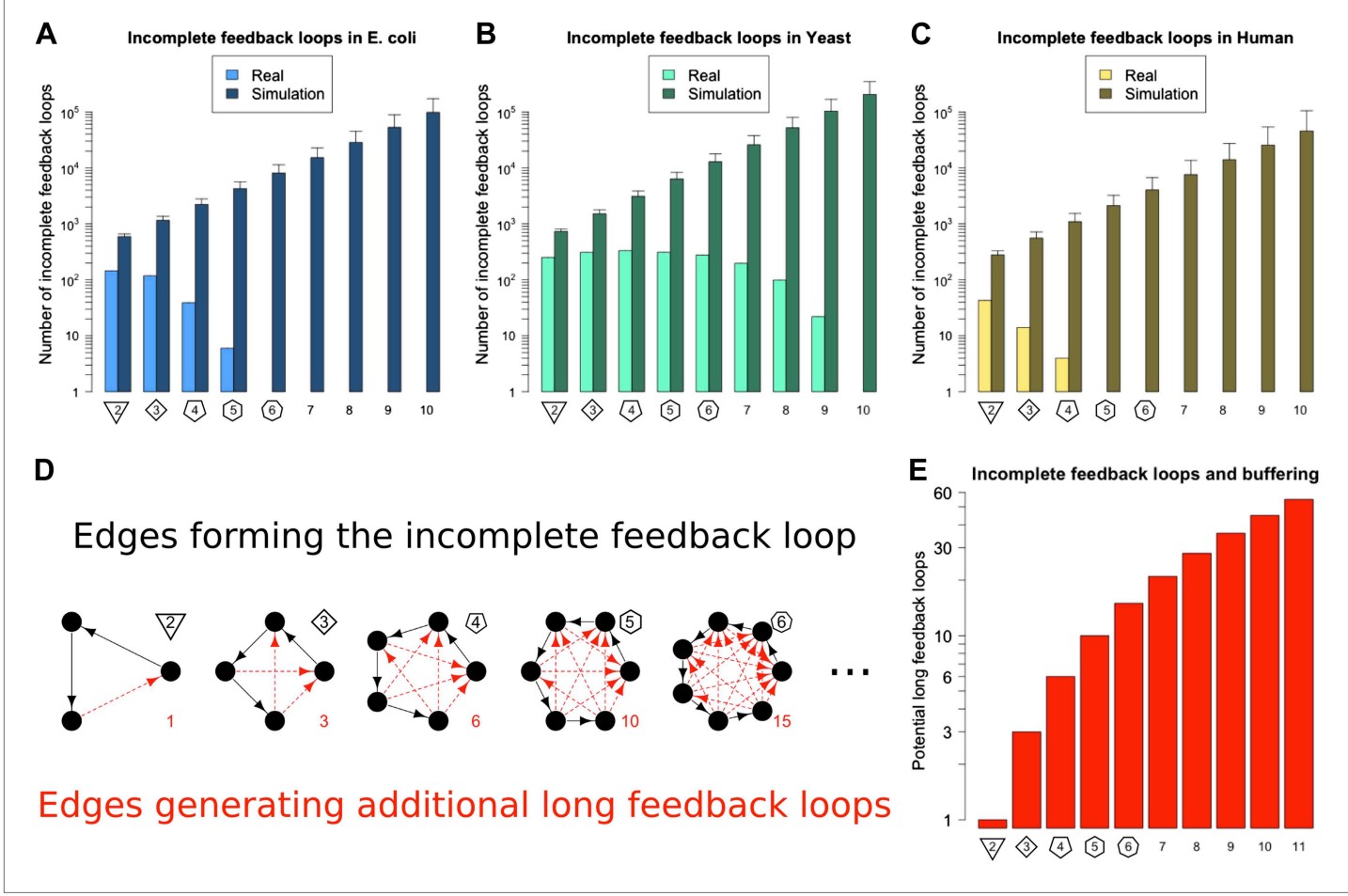

**Figure 3**. Incomplete feedback loops in real and simulated GRNs. Number of incomplete feedback loops is provided on a logarithmic scale for *E. coli* (**A**), *S. cerevisiae* (**B**), and the human GM12878 cell line (**C**) and in each case is compared with a randomly simulated network containing the same number of genes, TFs, and connections. For the random networks, each graph reports the mean and standard deviation. (**D**) Schematic illustrations of incomplete feedback loops of length 2–6 are plotted in black, with the edge whose additions would result in additional long feedback loops plotted in red. (**E**) The number of long loops (>2 nodes) that can be created by the addition of a single new connection to incomplete loops of the specified length (numerical values also given in red text in panel **D**).

The following figure supplements are available for figure 3:

**Figure supplement 1**. Incomplete feedback loops in *P. aeruginosa*, *M. tuberculosis* and other yeast datasets.

**Figure supplement 2**. p-Value estimation for the number of long incomplete feedback loops.

**Figure supplement 3**. Number of incomplete feedback loops in the RegulonDB (*E. coli*) and Harbison et al. (yeast) datasets under different statistical conditions.

**Figure supplement 4**. Number of feedback loops under different random models for RegulonDB (*E. coli*).

**Figure supplement 5**. Incomplete feedback loop distribution in different networks.

*Figure 3—figure supplement 1A,B*, heavily shaded bars, and *Figure 3—figure supplement 4*), which on average have a thousand-fold greater number of incomplete feedback loops (>$10^5$) of a significantly larger size. Statistical analyses suggest that there is an extremely small probability (<$10^{-19}$) that the absence of long incomplete feedback loops is a chance event in the three organisms considered (*Figure 3—figure supplement 2A–C*). These results are relatively robust to variations in the confidence

levels of the *E. coli* and *S. cerevisiae* GRN (*Figure 3—figure supplement 3A,B*), and remain valid when different random models are considered (*Figure 3—figure supplement 4*). Note that the distribution of incomplete feedback loops is indicative of the different topological structures that can be observed in the network, and is not necessarily monotonically decreasing (*Figure 3—figure supplement 5*).

The striking difference between real and random networks in the preponderance of both the number of long feedback loops and incomplete feedback loops strongly suggests a profound selective pressure on living organisms to adopt GRN topologies that are stable under all parameter regimes. In addition, these topologies efficiently prevent random mutations from introducing possible sources of destabilization. Note that feedback loops involving more than three genes are also highly susceptible to the creation of additional long feedback loops making them strongly disadvantageous to stability. We say that networks configured to minimize the number of real and incomplete long feedback loops possess Buffered Qualitative Stability (BQS). Networks having this property are *stable* to perturbations and are also *buffered* against the potentially destabilising effects that occur when new links are added. If BQS is really a fundamental design principle of GRNs, as our data seem to suggest, they should display a range of other properties, which we describe and examine henceforth.

## BQS predicts intermediate-scale properties of GRNs

An important global network property constrained by BQS is the degree of cross-regulation between TFs. Since a TF must be both regulated and regulating to take part in a feedback loop, one way that GRNs could satisfy BQS and minimise the risk of unstable loops being formed, is by having a high proportion of TFs that are not regulated by other TFs. Consistent with this prediction, the percentage of unregulated TFs in *E. coli*, *S. cerevisiae*, *M. tuberculosis*, *P. aeruginosa,* human and other yeast datasets is very high (*Figure 4A*, *Figure 4—figure supplement 1A–E*). Comparison with random networks indicates that the probability of obtaining this proportion of unregulated TFs by chance is between $10^{-68}$ and $10^{-39}$ (*Figure 4—figure supplement 2A–C*). Similar results hold for *M. tuberculosis* (*Figure 4—figure supplement 1A*), *P. aeruginosa* (*Figure 4—figure supplement 1B*), and other yeast datasets (*Figure 4—figure supplement 1C–E*). These results are robust to variations in the confidence levels of the *E. coli* and *S. cerevisiae* GRNs (*Figure 4—figure supplement 4A,B*), and remain valid when different random models are considered (*Figure 2—figure supplement 4A*).

Some TFs, however, must be regulated by other TFs in order for the GRN to be able to combine information from multiple pathways and change state depending on different circumstances. In order to satisfy BQS, highly connected TFs should either be regulated by a large number of other TFs or should themselves regulate a large number of target TFs, *but not both* (since otherwise the TF in question is significantly more susceptible to becoming part of a 3-gene feedback loop after addition of a link); in other words, highly centralised control is disallowed. This prediction of BQS is indeed verified in *E. coli* (*Figure 4B*), *S. cerevisiae* (*Figure 4—figure supplement 3A*) and human (*Figure 4—figure supplement 3B*). The *E. coli* TF with the largest number of 'outgoing regulatory connections' regulates 38 other TF genes, but is itself regulated by only one TF; the TF with the largest number of 'incoming regulatory connections' is regulated by nine other TFs but itself regulates only one TF gene. There are no *E. coli* TFs that are both highly regulated and highly regulating (in fact, there are no TFs regulating >2 other TFs that are themselves regulated by >2 TFs).

BQS does not even allow central control to be split by connecting highly regulated TFs to highly regulating TFs, as this would create a large number of incomplete loops. Consistent with this idea, no *E. coli* or human TFs with in-degree >2 regulate TFs with out-degree >2 (*Figure 4C*, *Figure 4—figure supplement 3D*). In *S. cerevisiae* there are several TFs that exceed these limits, but this represents only a small minority of TFs (*Figure 4—figure supplement 3C*). These results show that BQS strongly favours distributed control over central control, and provide another example of BQS being a key determinant of the topology of GRNs.

## BQS predicts small-scale properties of GRNs

Our discussion so far has focused on the effect of BQS on large- and intermediate-scale global properties of GRNs. BQS also makes strong predictions about the small-scale local structure of GRNs. To investigate this, we dissected each of the GRNs into a series of small motifs comprising three or four genes (*Alon, 2006*; *Milo et al., 2002*). A single motif can, in principle, break Qualitative Stability by forming a feedback loop composed of three or more genes. As we have shown above, such motifs are essentially absent from real GRNs. However, motifs may be susceptible to feedback loop formation through the

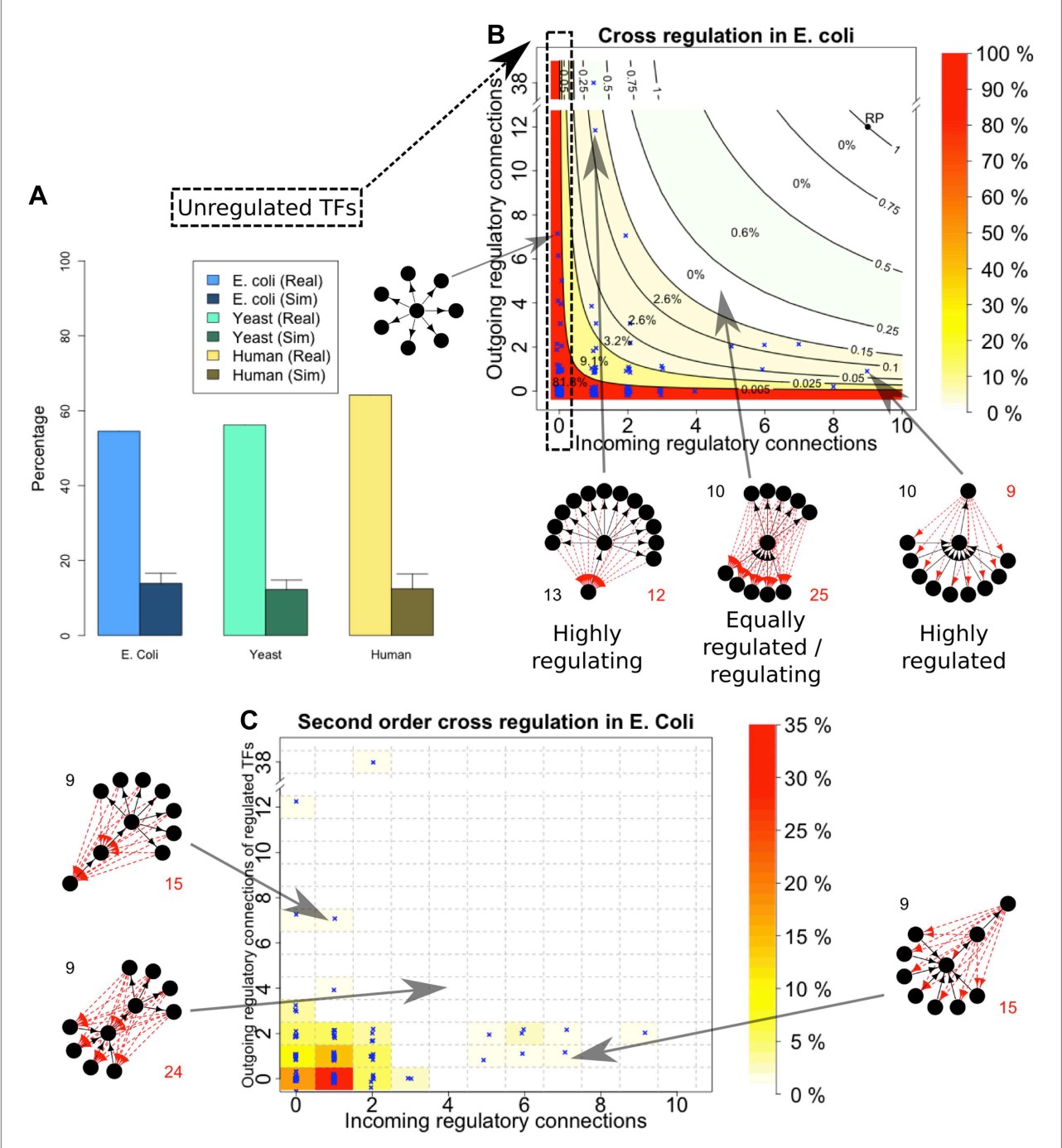

**Figure 4**. Evidence for BQS from TF regulation. (**A**) For each organism, the percentage of TFs which are not regulated by any other TF is shown. Comparisons are made to randomly simulated networks containing the same number of genes, TFs, and connections. (**B**) Each of the 154 TFs in the *E. coli* GRN is plotted in the space of incoming regulatory connections (number of regulatory links from other TFs) and outgoing regulatory connections (number of regulatory links to other TFs). Solid lines indicate isoclines of relative probability (normalized to unity at reference point RP) for creating a 3-gene feedback loop under random addition of a link. Percentages and colours indicate the fraction of TFs within each band demarcated by isoclines. (**C**) Each of the 154 TFs in the *E. coli* GRN is plotted in the space of incoming regulatory connections (number of regulatory links from other TFs) and outgoing regulatory connections of regulated TFs (number of regulatory links originating from a TF that is regulated by the selected TF). In (**B** and **C**), schematic motifs are provided; black solid arrows indicate the motif, while red dashed arrows indicate potential arrows whose addition results in the formation of long feedback loops. For each motif, the number of actual arrows is indicated in black and the number of potential destabilising arrows is indicated in red.

*Figure 4. Continued on next page*

*Figure 4. Continued*

The following figure supplements are available for figure 4:

**Figure supplement 1**. Unregulated TFs in *P. aeruginosa*, *M. tuberculosis* and other yeast datasets.

**Figure supplement 2**. p-Value estimation for the number of unregulated TFs.

**Figure supplement 3**. Cross regulation and second-order cross regulation in yeast and human.

**Figure supplement 4**. Number of unregulated TFs in the RegulonDB (*E. coli*) and *Harbison et al. (2004)* (yeast) dataset under different statistical conditions.

addition of a link, and we can therefore speak of 'buffered motifs' as motifs that are resilient to this, and therefore enhance BQS locally. Note that, to prevent possible biases introduced by the large number of non-TF genes, only motifs completely formed by TFs were considered. Using symmetry arguments, we grouped 3- and 4-gene motifs into buffered and non-buffered categories, which are equi-probable in a random network (confirmed by *Figure 5—figure supplement 4B,E,F,H,K,L*). *Figure 5A–F* show that in the real GRNs of *E. coli*, *S. cerevisiae* and human, buffered motifs (blue) are much more abundant than would be expected by chance, while unbuffered (green and violet) motifs are much less abundant; and indeed, unbuffered motifs which are particularly susceptible to breaking BQS (violet) are rare. Similar results hold for other confidence levels of *E. coli* (*Figure 5— figure supplement 3A–L*), other confidence levels of *S. cerevisiae* (*Figure 5—figure supplement 3M–X*), *M. tuberculosis* (*Figure 5—figure supplement 2A,B*), *P. aeruginosa* (*Figure 5—figure supplement 2,D*), and other yeast datasets (*Figure 5—figure supplement 2E–J*). Note that the IDs used in *Figure 5—figure supplements 2–4* are described by *Figure 5—figure supplement 1*.

These findings, besides confirming the role of BQS, provide additional support and potential explanations for the prevalence of well-studied motifs such as the feedforward loop (a stable motif) and the bi-fan (a buffered stable motif) as building blocks of networks (*Alon, 2006*; *Milo et al., 2002*). Indeed, the most buffered 3- and 4-gene motifs studied here are, respectively, latent feedforward loops and latent bi-fans.

## Measuring BQS

When a network uses only a subset of the possible edges in a motif, new connections can be added to expand the network (*Figure 5G*). In general, some of these connections will create long feedback loops (red dashed arrow), while others will not (green dashed arrow). As we have shown, robustness appears to exert a strong selective pressure on the topology of GRNs. To assess the extent of this pressure we used an extensive computational approach to estimate the probability that a random edge addition between two TFs creates a long feedback loop in the GRNs. All the possible edge insertions were tested in the real GRNs of *E. coli*, *S. cerevisiae* and human, and the values were compared with those estimated in the corresponding random networks. The extent of buffered stability is quite remarkable: 4363 new interactions can be added to the human GRN, but only 48 of them will create long feedback loops (*Supplementary file 1*). By contrast, addition of single extra links to random networks would lead to the creation of approximately 2000 different feedback loops on average: a hit rate of approximately 50%. The probability that real GRNs will gain long feedback loops by random edge additions is low (*Figure 5H*, lightly shaded bars), and is significantly smaller than that found for comparable random networks (*Figure 5H*, heavily shaded bars). Nevertheless, these probabilities are non-zero, indicating a trade-off between stability and the need for cells to regulate gene expression.

## BQS highlights critical network modules

Qualitative Stability is compromised to a very small degree in the GRN of *E. coli* by the two small, but 'illegal' feedback loops shown in *Figure 6A,B* and highlighted in *Figure 6—figure supplement 1*. Three aspects are noteworthy. Firstly, of the seven 2-node feedback loops in the *E. coli* GRN, four are embedded into the two potentially unstable motifs depicted in *Figure 6A,B*, whilst the other three are isolated from other feedback loops and so are either stable or likely to act as switches. Secondly, the genes comprising the two illegal motifs are involved in drug resistance (*Ariza et al., 1995*; *Martin and Rosner, 2002*; *Nishino et al., 2008*; *Keseler et al., 2011*; *Figure 6C*) and/or acid resistance (*Sayed et al., 2007*; *Keseler et al., 2011*; *Figure 6C*). This raises the possibility that limited instances of deviation from

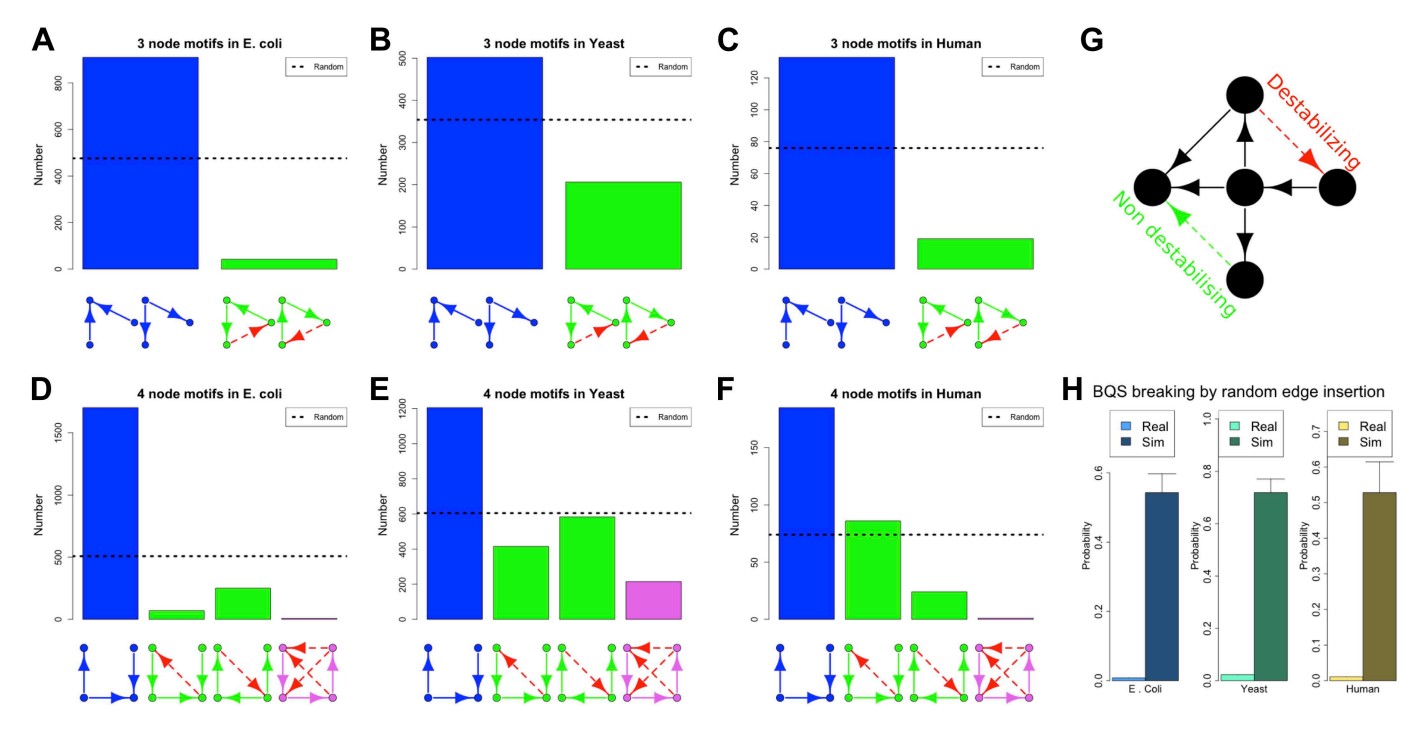

**Figure 5**. BQS in selected 3- and 4-gene motifs. Count of 3- and 4-gene motifs in the GRNs of *E. coli* (**A** and **D**), *S. cerevisiae* (**B** and **E**), and human GM12878 cell line (**C** and **F**) are classified into different categories. The y-axis reports the number of motifs plotted under each bar using blue (buffered), green (mono-unbuffered) or violet (poly-unbuffered) solid arrows. Red dashed arrows are not part of the motifs but indicate the additional links that would create a 3- or 4-gene feedback loop. The motifs selected have an equal probability of occurrence in a random network (See also *Figure 5—figure supplement 4B,E,F,H,K,L*). Horizontal black dotted lines in panels **A**–**F** indicate the motif abundance expected in random networks. (**G**) In the network schematic indicated by black arrows, adding a new connection can affect BQS; certain arrows will create a long feedback loop (red dashed arrow), while others will not (green dashed arrow). (**H**) Probability that a random edge addition between TFs would result in the creation of a long feedback loop is reported compared to random simulations. The different values observed in the simulations are due to the different number of TFs and regulatory connections in the three organisms.

The following figure supplements are available for figure 5:

**Figure supplement 1**. Labelling conventions for Figure Supplements.

**Figure supplement 2**. 3- and 4-gene motifs in *P. aeruginosa*, *M. tuberculosis* and other yeast datasets.

**Figure supplement 3**. 3- and 4-gene motifs in the RegulonDB (*E. coli*) and *Harbison et al. (2004)* (yeast) datasets under different statistical conditions.

**Figure supplement 4**. 3- and 4-gene motifs under different random models for RegulonDB (*E. coli*).

BQS may arise through evolution as a short-term expediency allowing survival in a changing environment. Thirdly, both loops share a remarkably similar sub-structure: two linked 2-gene feedback loops connected by one link into a 3-node feedback loop. It has been previously shown in chemical networks that such a configuration can display chaotic behaviour (*Sensse and Eiswirth, 2005*). It is tempting to speculate that these illegal motifs act as localized sources of chaos, allowing a cell population to quickly explore very diverse gene expression levels, thus accelerating the emergence of resistant phenotypes (*Lopez-Maury et al., 2008*). Moreover, compatible with the idea that chaotic behaviour should be tightly controlled, most of the genes depicted in *Figure 6A,B* are highly regulated (*Figure 6D*).

These ideas are also supported by the *M. tuberculosis* GRN: all the four genes involved in the formation of illegal motifs are implicated in stress responses (*He et al., 2006*; *Rodriguez et al., 2002*), and the two 3-gene feedback loops share the same topology observed in *E. coli* (*Figure 6—figure supplement 2A,B*). In addition, of the six 2-node feedback loops observed in the *M. tuberculosis* GRN,

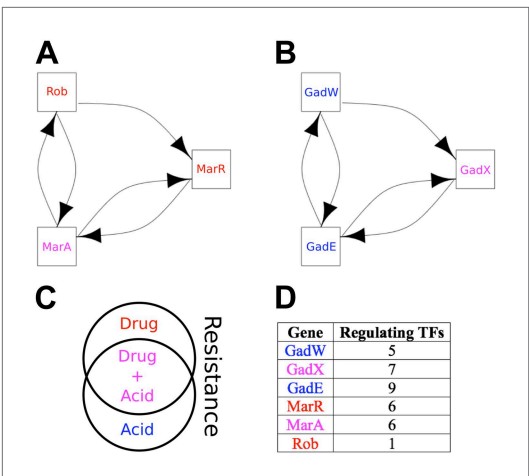

**Figure 6**. The two 'illegal' feedback loops in *E. coli*. The figure shows connections between TFs in the two motifs in *E. coli* that contain feedback loops with >2 links (**A** and **B**). Note the common feature of two adjacent 2-gene feedback loops. (**C**) Colours indicate if a gene is implicated in drug resistance (red), acid resistance (blue), or both (violet). (**D**) The number of regulatory connections for each TF in (**A** and **B**).

The following figure supplements are available for figure 6:

**Figure supplement 1**. The transcription factor GRN in *E. coli* with long feedback loops highlighted.

**Figure supplement 2**. Long feedback loops in *M. tuberculosis*.

**Figure supplement 3**. Long feedback loops in Lee et al. (yeast).

three are isolated from other feedback loops and the other three are embedded into the two potentially chaotic motifs. Finally, it is noteworthy that the 4-gene feedback loop in the *M. tuberculosis* GRN is formed by *joining* the two 3-gene feedback loops (*Figure 6—figure supplement 2C*), consistent with our earlier observation that long feedback loops are susceptible to the formation of additional embedded feedback loops.

*P. aeruginosa* has no long (>3 genes) feedback loops, and of its seven 2-node feedback loops five are isolated, and so are either stable or likely to act as switches, whilst the other two are linked but are of a form (positive-negative) that makes them Qualitatively Stable. Both of the *S. cerevisiae* 2-node feedback loops are isolated, whilst the human GM12878 cell line has only a single 2-node feedback loop which is therefore isolated. Curiously, the only long feedback loop observed in the yeast GRN derived by *Lee et al. (2002)* presents the same potentially chaotic topology discussed above. However, this illegal motif is not present in the more recent GRN derived by *Harbison et al. (2004)* (*Figure 6—figure supplement 3*).

## BQS is lost in cancer

Cancer cells have a dysregulated behaviour, breaking the 'social contract' necessary for maintenance of a healthy multicellular organism. Even within an individual tumour a wide range of cellular phenotypes is often observed (*Marusyk et al., 2012*). To some degree, this is likely to be a consequence of the genotypic heterogeneity of tumours. However, cancer cells also appear to be phenotypically less stable than normal cells (*Brock et al., 2009*; *Gupta et al., 2011*). Might this phenotypic instability result from a breakdown of BQS in cancer cells? There is currently only a single cancer cell line, the human leukaemia cell line K562, for which a high quality system-wide GRN has been derived (*Gerstein et al., 2012*). We therefore investigated the topological differences between the GRNs of K562 and the human non-cancer cell line GM12878. *Figure 7* and *Figure 7—figure supplement 1* show that the feedback loop distribution in the GRN of the leukaemia cell line is strikingly different from that of the non-cancer cell line. In K562, significantly more moderately long feedback loops of 4–8 genes are present (*Figure 7A,B*), and the number of loops formed by 3–5 genes is comparable to the number expected in a random network (*Figure 7A*). The number of incomplete feedback loops is also significantly larger in K562 (*Figure 7C,D*). In addition, poorly buffered motifs are abundant (*Figure 7E,F*) and the TFs cross regulation is less extreme (*Figure 7—figure supplement 2A,B*).

Interestingly, only a limited number of TFs contribute to the formation of the 59 long loops in the K562 GRN (*Figure 7G*) and it can be substantially 'stabilized'—that is most of the long feedback loops can be removed—by the removal of single pairs of TFs: FOSL1 and JUNB, JUNB and EGR1, or EGR1 and CEBPB. JUNB and FOSL1 are proto-oncogenes that are components of the AP-1 transcription complex (*Kouzarides and Ziff, 1989*) and have been reported to be part of a long feedback loop in ovarian cancer (*Stelniec-Klotz et al., 2012*); EGR1 is a regulator of tumour suppressor genes and an oncogene itself (*Baron et al., 2005*); and the TFs of the C/EBP family have been described as both tumour promoters and suppressors (*Nerlov, 2007*). These genes are also active in the non-cancer cell line GM12878, yet their destabilizing role in cancer arises from a rewiring of the network. GM12878 satisfies BQS whilst K562 comparatively does not, and yet the two GRNs have similar

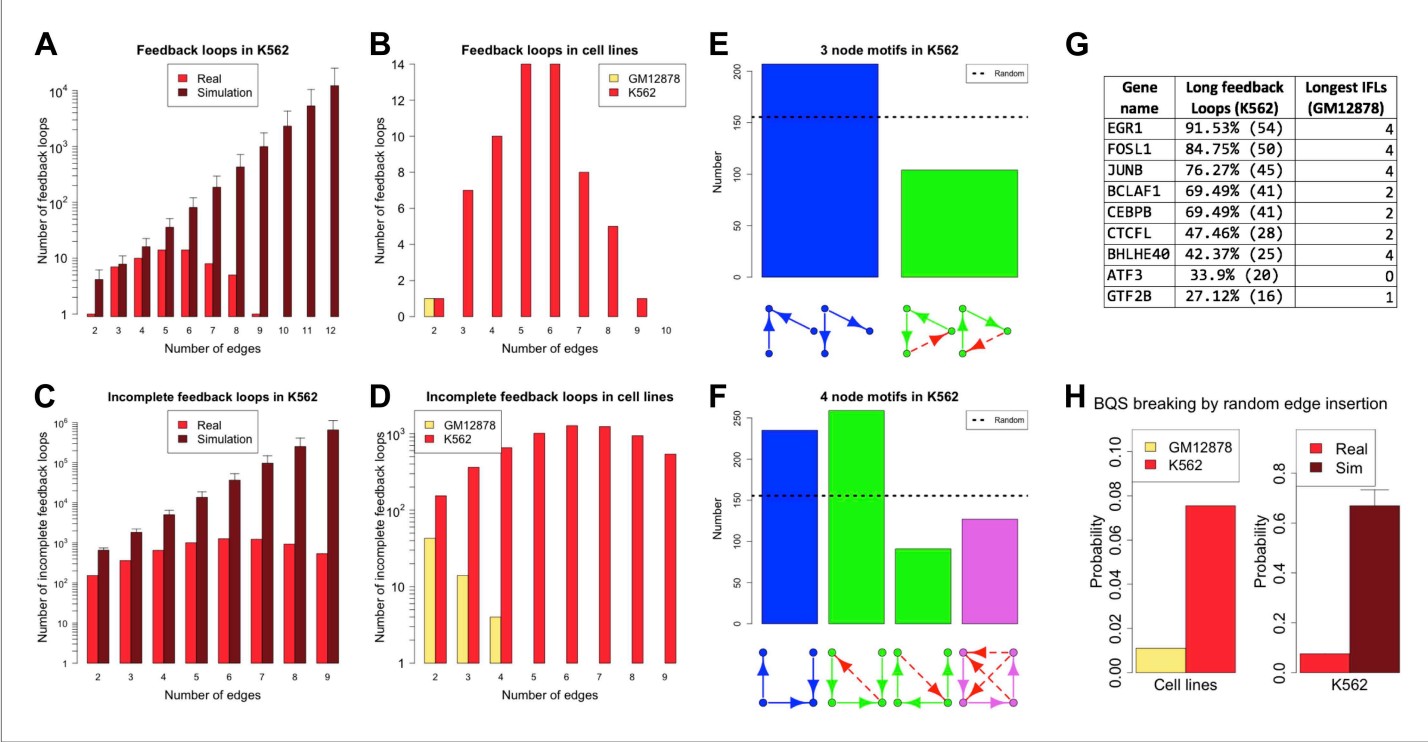

**Figure 7**. Broken BQS in a human cancer cell line. (**A** and **C**) Number of feedback loops and incomplete feedback loops in the GRN of the human cancer cell line K562, compared to the corresponding random network. Data is provided on a logarithmic scale. (**B** and **D**) Number of feedback loops and incomplete feedback loops in the GRN of K562, compared to the human non-cancer cell line GM12878. Note the use of a linear scale in (**B**). (**E** and **F**) Motif analysis of BQS for K562 for 3- and 4-gene motifs, using the same convention as described in *Figure 5*. (**G**) Genes implicated in the formation of long feedback loops in K562 are reported with the number and percentage of loop created, the length of the longest incomplete feedback loops involving the gene in GM12878 are also indicated. (**H**) The probability that a random edge addition would result in the creation of a long feedback loop is reported for both GM12878 vs K562 and K562 vs randomly simulated networks containing the same number of nodes and connections. Note that the GRNs of GM12878 and K562 have similar link densities. They comprise 4071 and 4049 genes respectively, and 8466 and 11,707 regulatory connections respectively.

The following figure supplements are available for figure 7:

**Figure supplement 1**. The transcription factor GRNs for GM12878 and K562.

**Figure supplement 2**. Cross regulation and second-order cross regulation in cancer.

**Figure supplement 3**. Longest incomplete feedback loops in the GM12878 GRN and their destabilization potential.

numbers of genes and regulatory connections (in fact, the K562 TF network has a lower link density than that of GM12878). Finally, the probability of introducing long feedback loops by random edge additions is significantly larger in the cancer cell line than in non-cancer cells, but still lower than the value expected in a random network (*Figure 7H*). These findings suggest that cancer cells break BQS and have a vastly less stable GRN than normal cells, which is however less unstable than expected in a random network.

Certain features of GRNs, such as rare, long incomplete feedback loops, make them more susceptible to the formation of long feedback loops when new regulatory connections are added (*Figure 7—figure supplement 3A–G*). Therefore, we investigated how many of the long feedback loops observed in K562 cancer cells could have originated from these relatively unbuffered network structures in non-cancerous GM12878 cells. We find that the three genes that contribute the most to the formation of long feedback loops in the cancer cell line (EGR1, FOSL1, and JUNB) are all involved in the formation of the longest incomplete feedback loops observed in the non-cancer cell line (*Figure 7G*). As highlighted earlier (*Supplementary file 1*), single insertions of a new connection into the non-cancer human cell line can create 48 different long feedback loops. Of these 48 potentially destabilising

interactions of the non-cancer cell line, three are actually observed in the cancer cell line, namely JUNB-BCLAF1, BATF-EBF1, and JUNB-EBF1. The likelihood of these potentially destabilising interactions occurring in the cancer cell line is 3/48 = 0.063. In comparison, there are a total of 4363 additional links that could be made between the TFs of the non-cancer cell line. Of these potential links, 56 are observed in the cancer cell line. Hence, the links observed represent 56/4363 (0.013) of all the possible ones. Therefore, the destabilizing interactions are five times more abundant than would be expected by chance (0.063 vs 0.013), consistent with the idea that destabilizing interactions were positively selected for during the microevolution process underlying cancer progression. This suggests that, despite the diverse biological histories of these two cell lines (the networks of GM12878 and K562 share only four common links between TFs), the progression of K562 into a cancerous state has involved changes to regions of the GRN in the progenitor normal cells that displayed weaker BQS. Such genetic factors are therefore likely to play a pivotal role in the process of cancer progression in other cell types.

## BQS and transcriptional response

The eukaryotic GRNs analysed so far are static 'snapshots' of potential transcriptional interactions of a population of cells under rich media growth. As discussed in 'Materials and methods', such conditions are ideal to minimize cell heterogeneity and to obtain high quality equilibrium networks that are ideally suited for theoretical analysis. However, during the lifetime of an organism GRNs are dynamic and the set of actual transcriptional regulations can change (*Luscombe et al., 2004*). GRNs transitioning from one transcriptional program to another are unlikely to be at steady state and, under these circumstances, transcriptional robustness may be less important. We decided to test if BQS could provide new insights into the role of robustness during such structural changes. To this end, we used the data presented in *Garber et al. (2012)* to reconstruct the GRN of murine dendritic cells at four different time points after stimulation by pathogens: at the time of stimulus application (marked as '0 hr') and after the cells have been exposed to the stimulus for 30 min ('0.5hr'), 1 hr ('1 hr'), and 2 hr ('2 hr'). Since the number of TFs studied in these networks is much smaller than the number considered previously, a different simulation algorithm has been used (see 'Materials and methods').

The '0 hr' GRN was obtained under conditions comparable with the organisms discussed previously. *Figure 8* shows that all the predictions of BQS are met at this time. As indicated by *Figure 8A*, the GRN has a very limited number of long feedback loops (only three with three or more genes), in striking contrast with the hundreds observed in randomly generated networks of the same link density. Interestingly, all of the long feedback loops depend on the transcriptional interaction between SFPI1 and E2F1 (*Figure 8—figure supplement 1A–C*). Notably E2F1 plays a crucial role in the cell cycle and is only transiently activated at commitment to cell division at the end of G1. Therefore, all of the long feedback loops detected are likely to be transient. Similarly, the number of incomplete feedback loops is very small, and much lower than would be expected in random networks (*Figure 8B*). Motif analysis is also consistent with BQS: there is a much higher proportion of unregulated transcription factors than would be expected by chance (*Figure 8E*), and the proportion of stable 3- and 4-node motifs is heavily biased towards the buffered stable forms that enhance BQS (*Figure 8C,D*). Additionally, the mode of cross regulation—with transcription factors tending to be either highly regulating or highly regulated (*Figure 8F*)—also follows the distribution predicted by BQS. Finally, the probability of creating additional long feedback loops in the network by randomly inserting a new regulatory connection is only 0.18, much lower than the value of 0.74 expected in a comparable random network.

Since a full eukaryotic transcriptional response typically requires >1 hr, we expect the '0.5 hr' network to be similar to the '0 hr' one. Indeed, the '0.5 hr' network still satisfies all the predictions of BQS and strongly resembles the '0 hr' network (*Figure 9A,B,G*). In contrast, at 1 and 2 hr after the stimulus, a marked deviation from BQS is observed. A significant number of new long loops are created, peaking at 1 hr and declining slightly by 2 hr (*Figure 9C,E*). 22 long feedback loops remain at 2 hr, but interestingly all depend on the transcriptional interaction between RUNX1 and CEBPB. The probability of creating additional long feedback loops is noticeably larger at 1 and 2 hr than in the previous networks (*Figure 9G*). There is also a significant increase in the number of 4-node unbuffered motifs at 1 hr, though this does not persist in the '2 hr' GRN. However, some components of BQS remain unchanged in the stimulus response. Incomplete feedback loops still remain preferentially short (*Figure 9—figure supplement 1A,E,I*), TF cross regulation remains essentially unchanged (*Figure 9—figure supplement 1D,H,L*) and the numbers of unbuffered 3-node motifs (*Figure 9—figure supplement 1B,F,J*) and unregulated TFs (*Figure 9—figure supplement 1C,G,K*) remain low. Taken together,

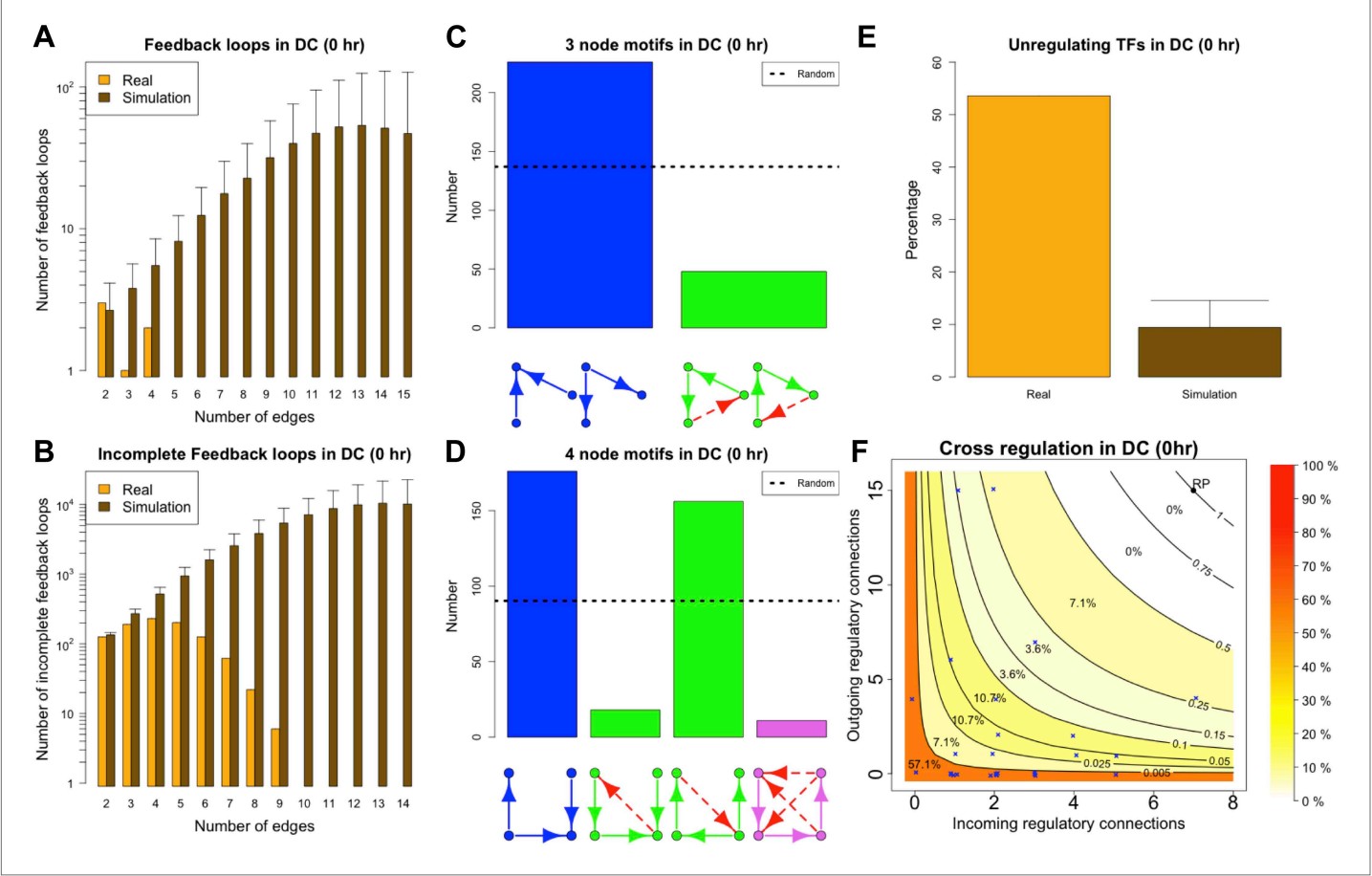

**Figure 8**. BQS in homeostatic murine dendritic cells. (**A** and **B**) Number of feedback loops and incomplete feedback loops compared to the corresponding random network. Data is provided on a logarithmic scale. (**C** and **D**) Motif analysis of BQS for 3- and 4-gene motifs, using the same convention as described in *Figure 5*. (**E**) Percentage of TFs which do not regulate other TFs compared to the corresponding random network. (**F**) Relation between the number of incoming and outgoing regulatory connections; the same conventions of *Figure 4B* are used.

The following figure supplement is available for figure 8:

**Figure supplement 1**. Long feedback loops in the GRN of homeostatic dendritic cells.

these observations show that in response to a stimulus that changes the transcriptional profile, there is a modest and probably transient loss of BQS as the GRN moves into a new configuration.

## Discussion

Previous work on GRNs, in addition to providing important insights into specific functionalities (*Kauffman et al., 2004*; *Albert, 2007*; *Karlebach and Shamir, 2008*), has highlighted some important common features at both local and global scales, in particular the prevalence of certain motif patterns (*Alon, 2006*; *Milo et al., 2002*) and scale-free degree distributions (*Strogatz, 2001*; *Barabasi and Oltvai, 2004*; *Albert, 2005*; *Buchanan et al., 2010*). Moreover, evidence of evolutionary pressures acting primarily on the topology of biological networks has been observed (*Tanay et al., 2005*; *Cross et al., 2011*). The biological principles and selective pressures underpinning the emergence of these characteristics are an active area of research (*Rosenfeld and Alon, 2003*; *Tyson and Novák, 2010*; *Liu et al., 2011*) and the identification of general principles is of pivotal importance for the progression of our understanding. By hypothesizing that GRNs must retain stability under a wide variety of circumstances, and so display Qualitative Stability, we provide a novel, simple and powerful explanation for numerous new and previously observed features of GRNs at different scales. We show that the GRNs of six different organisms display a remarkable lack of long feedback loops, which makes them Qualitatively Stable.

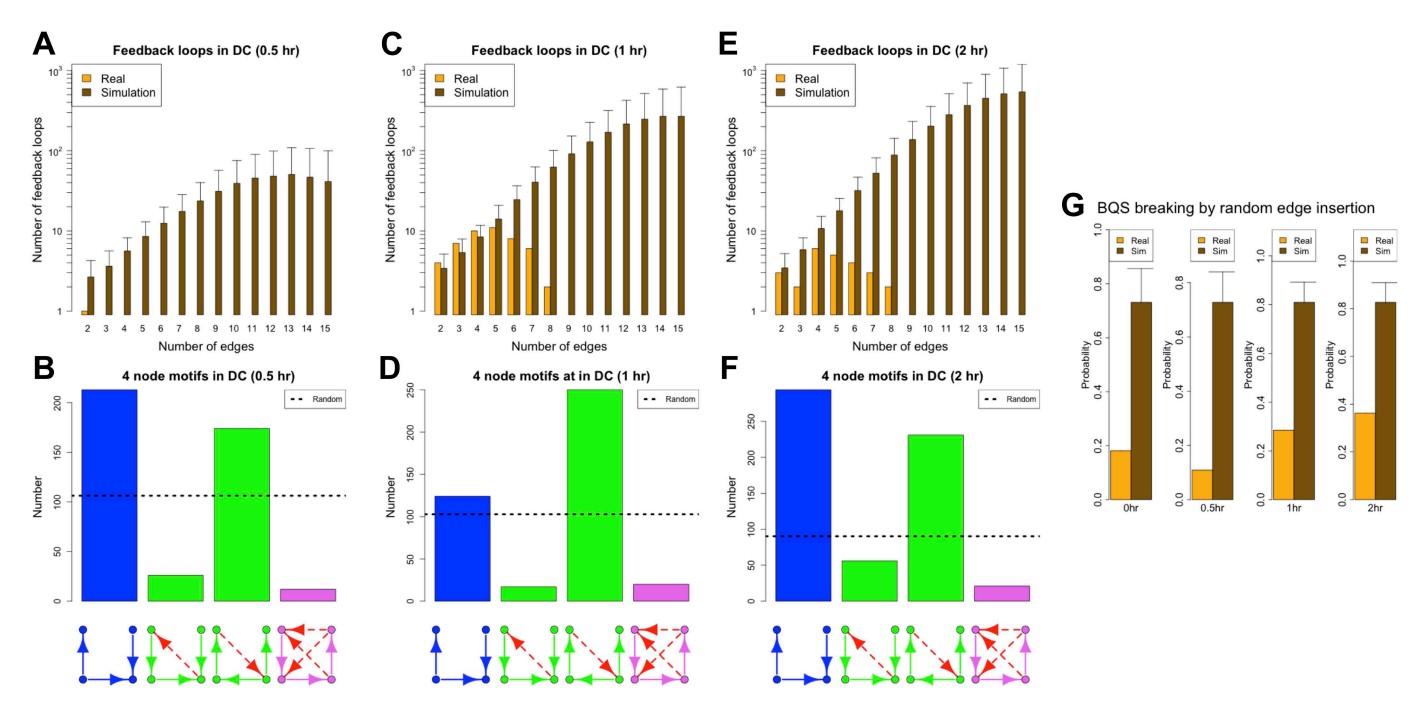

**Figure 9**. BQS during a transcriptional program activation in dendritic cells. (**A**, **C** and **E**) Number of feedback loops and incomplete feedback loops in the GRN of dendritic cells at different times after a stimulus compared to the corresponding random network. Data is provided on a logarithmic scale. (**B**, **D** and **F**) Motif analysis of BQS for 4-gene motifs, using the same convention as described in *Figure 5*. (**G**) Probability of creating additional long feedback loops by the addition of a random link between TFs in the GRN of dendritic cells at different times after a stimulus and the corresponding random network.

The following figure supplement is available for figure 9:

**Figure supplement 1**. BQS in the GRN of dendritic cells.

This means that perturbations in the strength of any individual interaction—the extent to which a TF activates one of its target genes—will not disturb the state of the network. Thus, GRNs are robust to a very wide range of perturbations. Indeed, the selective pressure for Qualitative Stability appears to be so strong that GRNs are heavily buffered to retain this property under the random addition of new network connections, a property we term Buffered Qualitative Stability (BQS). BQS is revealed in numerous different features of GRNs: the lack of long incomplete feedback loops, the high proportion of unregulated TFs, the lack of TFs that are both highly regulated and highly regulating, and the preponderance of buffered over unbuffered motifs. As well as providing stability to immediate disturbances (such as insults that cause a TF to regulate a promoter that it does not normally interact with), BQS also provides robustness to genetic changes that occur as a consequence of sexual recombination of alleles and to mutations that occur over evolutionary time. BQS therefore enhances the evolvability of GRNs, allowing them to function reproducibly in different genetic and environmental contexts. It is worth noting that the scarcity of feedback loops in *E. coli* has been remarked before (*Shen-Orr et al., 2002*), but the selective pressures underpinning this fact were not explored.

It is important to consider whether the potential under-sampling of GRNs (i.e., false negatives) and the inherent noise in data generation lead to an underestimation of the true link density in the networks, thus jeopardising the strength of our conclusions that networks satisfy BQS. However, defects in the GRN data are very unlikely to undermine our conclusions, for various reasons. First, as we have clearly shown, random networks with the same link density as the real GRNs do not satisfy BQS, whereas the real GRNs do. Second, as illustrated thoroughly in *Figure 2—figure supplement 3A–J*, *Figure 3—figure supplement 3A,B*, *Figure 4—figure supplement 4A,B*, and *Figure 5—figure supplement 3A–X*, the compliance of real GRNs to BQS is essentially preserved when the rates of

false positives and negatives are varied in the *E. coli* and *S. cerevisiae* GRN datasets. Third, the non-cancer (GM12878) and cancer (K562) cell lines show very different BQS properties, despite having similar link densities. Since the data from both sets of cells were derived under similar conditions and with the same algorithmic tools, they would presumably have similar rates of false positives and negatives. A similar comparison can also be made between the stimulated and unstimulated dendritic cells.

Robustness is a key feature of cellular behaviour, but an appropriate response time is also critical. Changes in the transcriptional profile of cells are generally slow, requiring from tens of minutes to hours (*Alon, 2006*). As a consequence, it may be unhelpful for a cell to propagate transcriptionally a signal in a long cascade, and this might also explain the limited number of long incomplete feedback loops observed. While it is likely that timing effects play a role in the organization of GRNs, there are several reasons for believing that robustness is still fundamental. First of all, certain transcriptional changes take place over very long time scales, indicating that a fast response time is not always necessary. Moreover, each TF in a transcriptional cascade is potentially post-transcriptionally controlled (e.g., from signalling pathways), and therefore the signal need not proceed linearly from the top to the bottom of the pathway, but instead genes at the bottom of a cascade could be activated independently. Additionally, it should be remembered that BQS still allows a long pathway (for example A-B-C-D-E) to have a short link between the start and end (A–E in this example); when we enumerated incomplete feedback loops we considered all the possible transcriptional paths through the network, whilst a fast transcriptional propagation may only involve the shortest path. Finally, we note that several key features of BQS that we observe in GRNs, including the prevalence of buffered 3- and 4-node motifs, the lack of transcriptional hubs with similar numbers of incoming and outgoing links, and the high prevalence of unregulated TFs are independent of signaling pathway length.

While our analysis suggests multiple possible connections between the breaking of BQS and the cancerous state of the K562 cell line, it is worth pointing out that K562 was derived from a stem cell population. This raises the possibility that some amount of the breaking of BQS observed may be connected with the K562 cell line's original stemness. Further experimentation on the robustness of other cell lines will allow to assess the aetiology of the lack of robustness observed, and whether the loss of BQS is associated to stem-like properties of cells.

Although BQS has been developed here in the context of GRNs, it provides general principles that can be used to analyse or manipulate robustness in any regulatory system, biological or otherwise. These principles are summarised by five simple rules (*Figure 10*): 1. Avoid long feedback loops to minimize instability arising from perturbations in network interactions (the basic principle of Qualitative Stability); 2. Favour constitutive (unregulated) nodes to reduce the potential number of loops; 3. Avoid long paths to minimize the number of 'incomplete feedback loops' and the emergence of instability due to addition of new network connections; 4. Favour buffered motifs over unbuffered motifs to reduce the potential number of loops; 5. Avoid centralized control hubs with a large number of both regulatory and regulating connections to reduce potential instability (necessitating the use of distributed control). These rules can be used to devise highly stable networks that minimize the 'hyper-risk' inherent in global networks that are difficult to control (*Helbing, 2013*).

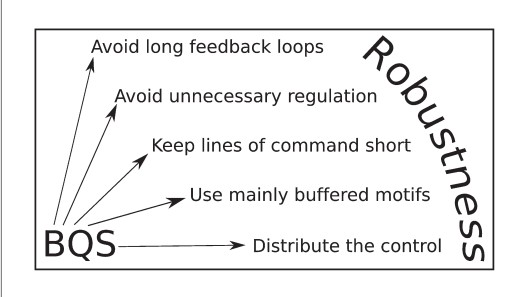

**Figure 10**. BQS Rules. BQS provides five general design principles applicable to any regulatory system. The rules are described in more detail in the 'Discussion' Section.

## Other biological networks

It is widely believed that feedback plays a major role in biological control (*Harris and Levine, 2005*; *Tsang et al., 2007*; *Peter et al., 2012*). As we have demonstrated here, BQS demands that GRNs are free of long feedback loops, even under the addition of new links. The question remains, to what extent feedback operates in post-transcriptional regulation rather than the purely transcriptional networks we have examined here. It was not possible for us to perform a similar analysis of post-transcriptional regulation because strongly validated system-wide data describing such interactions are not currently available. However, the biological literature provides

some clues. Motif analysis of post-transcriptional networks reveals that BQS-compliant feedforward loops are overrepresented, while BQS-breaking feedback loops are not (*Gerstein et al., 2010*; *The modENCODE Consortium et al., 2010*; *Cheng et al., 2011*; *Joshi et al., 2012*). This suggests that Qualitative Stability may still be an important principle governing the topology of these networks. Additionally, it has been noted that stable motifs are more common than unstable ones even in post-transcriptional signal transduction (*Milo et al., 2002*), suggesting again a role for Qualitative Stability in these networks. Nevertheless, the existence of feedback loops in post-transcriptional networks (*Harris and Levine, 2005*; *Tsang et al., 2007*) supports the idea that the severe constraints of BQS are 'loosened' to allow more responsive dynamic functionalities in post-transcriptional regulation. Consistent with these expectations, short feedback loops involving two or three genes appear to be present in some developmentally regulated gene networks (*Peter et al., 2012*). This raises the hypothesis that the different levels of gene regulation (transcriptional vs post-transcriptional) provide a way of segregating control modules with different robustness properties.

## Implications for active transcriptional response

The robustness provided by BQS allows a transcriptional network to filter out internal or external disturbances. Such robustness is desirable under normal conditions, but can be detrimental during a transcriptional response that requires effective and fast changes in the set of transcribed genes. Consistent with this idea, our analysis indicates that a certain level of instability builds up during a response to a stimulus that produces a transcriptional response. Quite remarkably our results also suggest that two hours after such a stimulus the GRNs considered are 'on the verge of stability', as the deactivation of just one transcriptional interaction will make the network robust according to the BQS rules. It is worth pointing out, however, that each cell in a population is responding to a stimulus independently and potentially at a different pace. Therefore, the apparent post-stimulus instability observed may be the result of sampling transcriptional occupancy of cells at different stages of the transcriptional response. Future experimentation focused on the robustness property of cells and single cell GRN reconstruction will likely clarify which interpretation is correct.

## Implications for evolvability and drug design

The molecular bases of evolutionary innovations are complex and poorly understood (*Wagner, 2011*). It is notable that the only instances where the *E. coli* and *M. tuberculosis* GRNs deviate from BQS are found in genes functionally related to stress responses. This might provide *controlled instability* allowing bacteria to explore new gene expression levels in response to environmental stresses, thus achieving a short-term evolutionary advantage. While other mechanisms are likely to be in play to achieve longer-term adaptations, therapeutic targeting of unstable motifs may provide a novel systems approach to drug discovery.

The deviation of the human cancer cell line K562 from BQS is also very striking. This deviation allows a cancer cell, in principle, to readily change its phenotype in response to internal or external stresses, so it can explore different phenotypic states, which might help its proliferation in otherwise challenging tissue environments, or even to survive drug treatment. Nevertheless, the cancer cell line still possesses a degree of BQS far greater than that observed in a random network. This is consistent with the role of the unstable motifs in *E. coli* and *M. tuberculosis* and supports the idea that a small breakdown of BQS might be a hallmark of recent or rapid selection pressure. We note that single-cell experiments report large changes in protein abundance occurring in individual cancer cells after drug treatment, consistent with our theory (*Cohen et al., 2008*).

Taken together, our results indicate that BQS adds a powerful new weapon to the arsenal of network medicine (*Barabasi et al., 2011*; *Noh et al., 2013*; *Pe'er and Hacohen, 2011*). The discovery that the transcriptional interactions most critically involved in the formation of long feedback loops in the cancer cell line K562 are also present in the longest incomplete feedback loops in the non-cancer cell line GM12878 may provide clues to mechanisms of cancer initiation and progression. Our theory suggests that random perturbations to the GRN of normal cells will probably leave its stability unchanged. Destabilization of the GRN is likely to involve those genes that are on the edge of stability, for example genes participating in long incomplete loops. It is striking that the two gene families with the highest *destabilization potential* in the non-cancer cell line (JUN and FOSL) are actually found to be the two gene families with the highest *destabilization role* in the cancer cell line. These observations are consistent with the notion of cancer as a 'systems disease' that involves changes that lead to

destabilisation of the GRN. If network instability is a general feature of cancer cell GRNs, analysis of this kind can potentially be used to design new anti-cancer strategies that exploit the unique weaknesses of cells lacking BQS.

## Materials and methods

### Conditions for Qualitative Stability of networks and its applicability to GRN

The study of stability in qualitative networks of interacting entities was introduced in the seminal work of Quirk and Ruppert in economics (*Quirk and Ruppert, 1965*). Since then, the idea has been applied to different disciplines (*May, 1973b*; *Tyson, 1975*) and the theory has been slightly enhanced (*Jeffries, 1974*). It has also been studied within the broader field of qualitative matrix theory (*Maybee and Quirk, 1969*; *Hale et al., 1999*). While the theory developed by Quirk and Ruppert is probably the most well-known tool to study Qualitative Stability, a similar formalism is provided by the work of Puccia and Levins on loop analysis (*Puccia and Levins, 1985*). The theory deals with systems at equilibrium and considers the effect of small perturbations. Stable systems are characterized by the ability to react to these perturbations by returning to their original equilibrium state. In qualitative networks, each node is associated with a quantity or concentration and the presence of an arrow from node A to node B indicates that changing the concentration of A has an effect on B. The sign of the arrow indicates whether increasing A increases (positive arrow) or decreases (negative arrow) B. The absence of an arrow indicates that increasing the concentration of A does not directly affect the concentration of B. Note that self-regulation, represented by an arrow from a note to itself, is also possible.

In its original form, the theory states four conditions for qualitative stability:

1. Absence of positive self-regulation
2. Absence of double positive or double negative two-node feedback loops
3. Absence of feedback loops longer that two
4. Invertibility of the sign matrix

Condition 1 prevents unlimited autocatalysis. Condition 2 prevents unlimited 'collaborative autocatalysis' (double positive) and switches (double negative), *but note that positive/negative two-node feedback loops are allowed*. Condition 3 is less intuitive, and disallows feedback systems that may go 'out-of-sync' for example as a consequence of non-linearities in the interactions. Condition 4 forbids the presence of two or more nodes that are being affected, or affect, the same nodes in the same way.

While conditions 1, 2 and 4 are very important from a theoretical point of view, they are of limited relevance to our analysis. For autocatalytic positive self-regulation, saturating effects on transcription (for example due to limiting numbers of RNA polymerases) along with protein degradation means that autocatalytic TFs will reach a stable steady-state rather than increasing to arbitrarily high values. Therefore condition 1 is of questionable relevance to GRNs. This argument of finite resources obviating positive self-regulation has also been applied to the study of Qualitative Stability in ecosystems (*May, 1973a*).

GRNs breaking condition 4 have no biologically plausible incarnation. In order for this condition to be violated, two different TFs must act on the network in the same way, that is, they need to regulate each other and promote and inhibit exactly the same set of genes, as this would make the columns of the sign matrix linearly dependent and therefore the matrix non-invertible. If two TFs act in the way described, they would be biochemically indistinguishable. Due to the methodology used to derive GRNs, two such genes would be collapsed into the same node. Condition 4 is therefore trivially verified.

The case of double positive and double negative feedback loops is more complex. Under biological constraints relevant for GRNs, isolated double negative feedback loops (i.e., not connected with other feedback loops formed by two or more nodes) are unstable only to the extent that they form switches that can exist in one of two stable states. Double positive feedback loops are potentially capable of a more complex behaviour. However, when considered in isolation with negative self-regulation under constraints relevant to GRNs, they are likely to display a switch-like behaviour (*Banerjee and Bose, 2008*). Remarkably, in both GRNs for which sign information is available these conditions are verified: in *P. aeruginosa* the only double positive feedback loop is isolated and in *E. coli* all non-isolated two-gene feedback loops are part of the potentially chaotic motifs discussed above, and are therefore not relevant for these particular 2-node stability arguments. Hence, the only scenario in which condition 2 potentially threatens Qualitative Stability in GRNs is a 'daisy chain' of linked 2-node feedback loops; in

the single case where such a daisy chain is observed (in *P. aeruginosa*) both 2-node feedback loops are of the positive/negative type and so result in a form compatible with Qualitative Stability. Taken together, the above considerations indicate that condition 3 is the most significant in a rigorous comparative study on the GRNs.

The introduction of a time delay into a system generally leads to an increased dimensionality. It is therefore not surprising that a delay introduced into feedback loops can lead to oscillations and instability, and this is one additional reason contributing to feedback loops containing >2 nodes not being Qualitatively Stable. Gene expression is a complex multi-step process and the role of regulatory mechanisms as potential sources of delay is an active area of research (*Gorgoni et al., 2014*). The complexity of the problem, and the difficulty in obtaining quantitative information, limit our ability to assess the role of delays on our model. A delay that is fast compared to transcriptional regulation can be ignored due to the separation of the time-scales. When this is not the case, delays are potential sources of instability, especially in the presence of feedbacks of any size. Delays may be stronger in eukaryotes than in prokaryotes, as transcription and translation are spatially separated. This may partially explain why the GRNs of yeast and GM12878 have a remarkably limited number of feedback loops of any size.

A final aspect of the theory is worth noting. The theory of Qualitative Stability has been developed by means of differential equations and the conditions discussed above properly apply only to deterministic autonomous systems. This results in our model being a simplification when compared to biological networks such as GRNs. It has been observed that cells limit the noise in gene expression (*Raj and van Oudenaarden, 2008*), suggesting the existence of biological mechanisms that reduce the extent of stochasticity. Therefore, whilst our model is likely to be largely compatible with the biological behaviour, noise may play a role in triggering GRN reorganization in response to strong stimuli.

It also is worth stressing that while stability is generally of pivotal importance, particular situations may require a fast, rather than stable, response. Therefore, when GRNs must be able to undergo changes of state, for example during development (*Peter et al., 2012*), or must be able to respond to external stimuli, for example to mount an immune response (*Ciofani et al., 2012*), the role of stability may be more limited.

## A brief review of available datasets and a rationale for those chosen

The reconstruction of the GRN of an organism is a challenging task, and an active field of research (*Kim and Park, 2011*). Different methodologies have been developed and each of them presents advantages and disadvantages. Nevertheless, certain techniques, such as PCR and ChIP-Seq, are generally regarded as more reliable.

Besides the technical problems, additional complications arise from the intrinsic working of gene interactions. The GRN is dynamic and changes according to external and internal conditions (*Harbison et al., 2004*; *Luscombe et al., 2004*). The sources of this variability are probably diverse and additional work will be needed to assess the cellular mechanisms that shape the GRN.

Moreover, the different molecular responses observed at a single-cell level (*Cohen et al., 2008*; *Tay et al., 2010*) suggest that the different stages of the cell cycle and different stress conditions may result in structurally different GRNs. Single-cell GRN reconstruction is currently beyond experimental reach, and stress is known to promote intra-population diversity (*Cohen et al., 2008*; *Lopez-Maury et al., 2008*). Therefore, where possible, we preferred to analyse GRNs obtained under rich media growth. These conditions result in a low cellular stress and thus are more likely to promote homogeneity in transcriptional response. This homogeneity minimizes the errors in GRN reconstruction due to *superpositions* of possibly different GRNs.

No GRN currently available should be expected to be a completely faithful representation of the real interactions among the genes. However, it is reasonable to assume that networks obtained with direct biological methodologies under controlled conditions are not biased towards certain topological features, and therefore provide a good representation of the topology of the real GRN.

Given these premises, it is not surprising that different GRNs are available in the literature for the same organism, and a choice had to be made to determine the datasets better suited for our analysis. We tried to select high quality datasets characterized by a statistical assessment of the interactions, a low rate of false positives, and public availability of the data.

The *Escherichia coli* RegulonDB dataset (*Salgado et al., 2012*) is probably the most validated GRN available in the literature. This dataset is regularly updated to incorporate new data, and is consistently

used as a basis for theoretical studies (*Alm and Arkin, 2003*; *Milo et al., 2004*). In its current state the dataset does not report the environmental conditions associated with each interaction. Therefore, it is likely that under any specific conditions only a subset of the interactions is active.

Recently, the GRNs of two other prokaryotes have been published: *Pseudomonas aeruginosa* (*Galan-Vasquez et al., 2011*) and *Mycobacterium tuberculosis* (*Sanz et al., 2011*). Since these organisms are less well studied than *E. coli*, their GRNs should be expected to be less complete.

Data on human GRNs are limited and the recent work by the ENCODE consortium (*Gerstein et al., 2012*) provided a unique opportunity to compare the GRNs of a non-cancerous and a cancerous cell line, studied under similar experimental conditions. Moreover, the methodology used to construct these networks (ChIP-Seq) and the carefully engineered protocol suggest a high degree of biological reality.

The situation for yeast is more complex. After the work by *Lee et al. (2002)*, other datasets have been made available. To the authors' knowledge, the work by *Harbison et al. (2004)* is the most exhaustive GRN derived from direct biological methods under stable conditions (rich media), and therefore the ideal workbench for a topological analysis.

Other yeast GRNs have been published and used for different types of studies about the genetic bases of yeast behaviour; in this context, *Luscombe et al. (2004)* and *MacIsaac et al. (2006)* are interesting examples.

*Luscombe et al. (2004)* extended *Lee et al. (2002)* by introducing additional interactions obtained under different environmental conditions, but the data available do not provide a statistical assessment for the interactions. Moreover, the data from *Garber et al. (2012)* raise the possibility that the GRN of yeast is different under different conditions, suggesting that the network derived by *Luscombe et al. (2004)* may not provide a faithful representation of an equilibrium yeast GRN. *MacIsaac et al. (2006)* used the data provided by *Harbison et al. (2004)* to construct a regulatory network encompassing different *Saccharomyces* species, and derived the more conserved interactions.

To the authors' knowledge, the dataset discussed in *Garber et al. (2012)* is the only one available where the author used ChIP-Seq to study the dynamics of transcriptional response. Other authors either make use of gene expression data—thus removing the purely transcriptional nature of the network—or consider only a handful of transcription factors—thus making the statistical analysis discussed here inappropriate. As remarked above, the heterogeneity of the transcriptional response in a population of cells is likely to contribute to the experimental error in this this type of data, and additional experimentation is important to verify our conclusions.

## Derivation of the networks and statistical analysis

The *E. coli* GRN was constructed using version 8 of the RegulonDB (*Salgado et al., 2012*) available at http://regulondb.ccg.unam.mx/. The network was restricted to those interactions supported by at least two evidence codes. The validity of our approach with a different number of evidence codes is assessed in *Figure 2—figure supplement 3A–E*, *Figure 3—figure supplement 3A*, *Figure 4—figure supplement 4A*, and *Figure 5—figure supplement 3A–L*.

The *S. cerevisiae* GRN was constructed using the interactions reported by *Harbison et al. (2004)* under rich media growth (http://younglab.wi.mit.edu/regulatory_code). The network was restricted to those interactions with a p-value lower than $10^{-3}$. The validity of our approach with different p-values is assessed in *Figure 2—figure supplement 3F–J*, *Figure 3—figure supplement 3B*, *Figure 4—figure supplement 4B*, and *Figure 5—figure supplement 3M–X*.

The human non-cancer and cancer cell GRNs were constructed using the filtered data constructed by *Gerstein et al. (2012)* for the GM12878 and K562 cell lines respectively. As previously observed, cells from different tissues generally display different GRNs (*Pe'er and Hacohen, 2011*; *Bensimon et al., 2012*). Therefore, we analysed the GRNs from the two cell lines separately. These networks are encoded by the files *enets8.GM_proximal_filtered_network.txt* and *enets7.K562_proximal_filtered_network.txt* respectively. The files are available at http://encodenets.gersteinlab.org/.

The murine dendritic cell GRNs were constructed using the interactions reported by *Garber et al. (2012)* available at http://www.weizmann.ac.il/immunology/AmitLab/data-and-method/iChIP/data. Only the interactions between transcription factors were considered. An edge is inserted from gene A to gene B at time T if

1. The protein product of A binds to a promoter area of B
2. The combined score for the binding is larger that 26.9
3. The score for the binding at time T is larger than 26.9

Note that the threshold value of 26.9 was extracted from the experimental procedure of *Garber et al. (2012)*.

The *P. aeruginosa* GRN was constructed from the dataset provided by *Galan-Vasquez et al. (2011)*. No filtering was applied and all the interactions were considered; therefore a perceivable level of false positives and negatives is to be expected.

The *M. tuberculosis* GRN was constructed from the dataset provided by *Sanz et al. (2011)*. The dataset includes a list of evidence codes for each interaction. However, most interactions are supported by only one evidence code. Therefore, no filtering was applied and all the interactions were considered. Similarly to *P. aeruginosa*, a perceivable level of false positives and negatives is to be expected.

The yeast dataset provided by *Luscombe et al. (2004)* does not include any systemic information on the statistical validity of the interactions. Therefore no filtering was applied and a perceivable level of false positives and negatives is to be expected.

The yeast dataset provided by *Lee et al. (2002)* was treated in the same way as *Harbison et al. (2004)* and only interactions supported by a p-value lower than $10^{-3}$ were considered for the general analysis.

To provide compatibility with the statistical conditions used in *Harbison et al. (2004)* and *Lee et al. (2002)*, the yeast dataset provided by *MacIsaac et al. (2006)* was constructed using the file *orfs_by_factor_p0.001_cons2.txt*. This file is available at http://fraenkel.mit.edu/improved_map/.

*Table 1* reports the number of genes, the number of transcriptional interactions and the network density for the full networks, identified by the (F), and for the networks composed by the transcription factor and the interaction among themselves, identified by (T), for all genome-wide datasets used in the article.

Feedback loops and incomplete feedback loops were computed by counting the number of sub-isomorphisms from the feedback or incomplete feedback loops for each GRN under consideration. For feedback loops this value was divided by the length of the loop to account for automorphisms. Note that, due to the nature of the analysis used (sub-isomorphism count), all the possible ways in

**Table 1.** Basic network properties for all the genome-wide GRNs used in the article

| Organism | Nodes (F) | Edges (F) | Nodes (T) | Edges (T) | Density (F) | Density (T) |
|---|---|---|---|---|---|---|
| *E. coli* | 1470 | 2909 | 154 | 140 | 0.0013 | 0.0059 |
| *M. tuberculosis* | 1624 | 3169 | 82 | 85 | 0.0012 | 0.013 |
| *P. Aeruginosa* | 692 | 991 | 85 | 81 | 0.0021 | 0.011 |
| Yeast (Lee) | 2417 | 4348 | 106 | 96 | 0.00074 | 0.0086 |
| Yeast (Harbison) | 2933 | 6152 | 169 | 188 | 0.00071 | 0.0066 |
| Yeast (Luscombe) | 3459 | 7053 | 142 | 254 | 0.00059 | 0.013 |
| Yeast (MacIsaac) | 2079 | 4097 | 116 | 134 | 0.00095 | 0.010 |
| GM12878 | 4049 | 11,707 | 82 | 84 | 0.00071 | 0.013 |
| K562 | 4071 | 8466 | 67 | 59 | 0.00051 | 0.013 |

Except when stated otherwise, random networks were generated preserving the number of transcription factors, genes, and interactions for *E. coli*, *P. aeruginosa*, *M. tuberculosis*, yeast, and the human cell lines. Random networks generated by preserving additional topological properties were also considered (See the subsection 'Effect of different constraints on the generation of random networks') and confirm the results discussed. For murine dendritic cells, random networks were constructed by preserving the number of transcription factors and interactions among them. Note that consequently the number of feedback loops and incomplete feedback loops is an underestimate with respect to the data considered in the other random networks. Self-regulating interactions were ignored. Since genes not encoding for TFs do not regulate other genes, a full-network analysis would *artificially increase* the stability property of the network. Therefore, to limit the bias introduced by the limited number of transcription factors, the number of incomplete loops, the motif abundance, the number of regulatory connections, and the probability of adding additional long feedback loops when a new regulatory connection is inserted were computed considering only the interactions among transcription factors. For each different type of random network, 1000 instances were generated for the data presented in the main Figures, while 100 instances were generated for the data presented in the Figure Supplements.

which a feedback loop or incomplete feedback loop can be created in the network are considered separately. Therefore, an edge or node is generally counted multiple times. The analysis is indicative of the general structure of the network, but can lead to counter-intuitive results.

Graphical motifs were computed in the usual way (*Milo et al., 2002*), but due to the different theoretical approach, no direct comparison with random networks was performed.

The probability of feedback creation by random addition of an edge in real GRNs was computed by trying all the possible edge insertions between the TFs and taking the ratio of insertions that form long feedback loops over the total number of insertions. This approach was computationally unfeasible for random networks and a sampling procedure was used: for each random network, 100,000 independent random insertions of one connection from two randomly selected transcription factors were tried and the probability of interest was estimated by considering the value:

$$\frac{N_S}{100000},$$

where $N_S$ is the number of insertions that resulted in the creation of long feedback loops. Additional details on the comparison of GM12878 and K562 are presented in *Supplementary file 1*.

Simulations and analyses were performed using R version 2.15.1 (*R Core Team, 2012*) and the 'igraph' package version 0.6-2 (*Csardi and Nepusz, 2006*).

## Effect of different constraints on the generation of random networks

It is common practice in statistics to use random simulations as a null model to test whether a feature can emerge with a high probability by chance. However, selecting the right type of randomness can be problematic. Network theory is no exception in this regard. Different constraints can be built into a simulation to obtain different types of random networks. To assess the role of different types of randomness on the properties of Qualitative Stability, we generated different types of random networks with different constraints using a variable number of characteristics of the *E. coli* network discussed in the main article.

We focused our attention on four characteristics of the real networks:

1. The number of genes (the number of vertices of the random network)
2. The number of interactions among the genes (the number of edges of the random network)
3. The number of transcription factors (the number of vertices allowing an out-degree larger than zero)
4. The absence of isolated genes (the absence of non-connected vertices)

Enforcing the number of vertices and edges of random graph is a common feature of random graphs: these random graphs are called Erdős–Rényi random graphs (*Bollobás, 2001*). However, it is less common in the literature to enforce additional characteristics. We stress that *including additional constraints results in a network that is less 'random'*.

The types of random networks that we considered are detailed in *Table 2*. The algorithms used to generate the different types of random networks have been implemented in R and are available as Source code. The functions used are listed by *Table 3*. Note that TF$^{Fixed}$IG$^{Not\ Allowed}$V1 and TF$^{Fixed}$IG$^{Not\ Allowed}$V2 use different algorithms to generate the networks:

- TF$^{Fixed}$IG$^{Not\ Allowed}$V1 generates an initial random network constructed by connecting each gene to one TF (selected at random). This ensures that no isolated genes are present. Then, if the number of edges used is less than the number required, edges are added at random until the expected number of edges is reached.

**Table 2.** Properties of the different random models used

| Model name | # of genes | # of interactions | # of TFs | Isolated genes |
|---|---|---|---|---|
| TF$^{Variable}$IG$^{Allowed}$ | Fixed | Fixed | Variable | Allowed |
| TF$^{Fixed}$IG$^{Allowed}$ | Fixed | Fixed | Fixed | Allowed |
| TF$^{Variable}$IG$^{Not\ Allowed}$ | Fixed | Fixed | Variable | Not allowed |
| TF$^{Fixed}$IG$^{Not\ Allowed}$V1 | Fixed | Fixed | Fixed | Not allowed |
| TF$^{Fixed}$IG$^{Not\ Allowed}$V2 | Fixed | Fixed | Fixed | Not allowed |

**Table 3.** Functions used to produce the different random networks

| Model name | Function used |
| --- | --- |
| TF$^{Variable}$IG$^{Allowed}$ | GenerateRandomNetwork.IG.Allowed() |
| TF$^{Fixed}$IG$^{Allowed}$ | GenerateRandomNetwork.IG.Allowed() |
| TF$^{Variable}$IG$^{Not\ Allowed}$ | GenerateRandomNetwork.IG.NotAllowed.V1() |
| TF$^{Fixed}$IG$^{Not\ Allowed}$V1 | GenerateRandomNetwork.IG.NotAllowed.V1() |
| TF$^{Fixed}$IG$^{Not\ Allowed}$V2 | GenerateRandomNetwork.IG.NotAllowed.V2() |

- TF$^{Fixed}$IG$^{Not\ Allowed}$V1 randomly adds edges between TFs and genes until a network with no isolated genes has been generated. At this point, if the number of edges used is less than the number required, edges are added at random until the expected number of edges is reached. Alternatively, if the number of edges is more than the number required, edges are removed at random, in such a way as not to create isolated genes, until the expected number of edges is reached.

TF$^{Fixed}$IG$^{Not\ Allowed}$V1 should be expected to be less biased, but is computationally extremely intensive, as a large number of edges usually needs to be removed. TF$^{Fixed}$IG$^{Not\ Allowed}$V2 uses a more biased procedure, but is more computationally tractable. As indicated by *Figure 2—figure supplement 4A,B*, *Figure 3—figure supplement 4*, and *Figure 5—figure supplement 4A–L*, the conclusions of the main text remain valid under all the conditions considered.

In our analysis we focused on a minimal number of constraints, in such a way to be able to assess the selective pressure of robustness at all scales. A detailed analysis of the adherence to BQS of different types of random networks is beyond the scope of this article and will be the subject of future investigations. However, given the existence of widely used more advanced models, it is important to justify our choice.

BQS provides predictions on GRNs at many scales, including degree distribution and motifs abundance, and the clear compliance of GRNs to such predictions indicates that robustness has left striking and detectable signatures. Therefore, using a method designed to preserve features such as degree distribution is likely to result in a network carrying the seed of robustness. To test this hypothesis, we assessed the effect of the degree-preserving model commonly used in motif analysis (*Milo et al., 2002*) for the *E. coli* GRN. This model, implemented by the function 'rewire' of igraph, reshuffles the original network in such a way to completely preserve the original in-degree and out-degree *for each single node*. Our results (not shown) indicate that feedback loops are relatively uncommon in degree-preserving random networks, regardless of the number of rounds of rewiring. However, incomplete feedback loops were more abundant and longer when compared to the real GRNs, even though to a lesser extent than purely random networks. Finally, the distributions of 3- and 4-node motifs were perturbed in such a way as to decrease the number of buffered motifs and to increase the number of unbuffered ones. However, the degree-preserving algorithm was unable to generate networks displaying an equal number of the 3- and 4-node motifs for the classes highlighted in the article, in stinking difference from a purely random model, even after 70,000 rounds of rewiring. Taken together, these observations support the idea that GRNs carry a strong signature of BQS and that random models constrained to be similar to GRNs will inherit, at least in part, many features of BQS. Our results also support the idea that purely random models may be the ideal null models to explore and highlight pervasive features of networks.

## Acknowledgements

The authors are grateful to Paul Campbell and Alessandro de Moura for helping to initiate this project. LA acknowledges partial support from the Human Frontier Science Foundation (RGP-0038). JJB acknowledges support from Cancer Research UK (grant C303/A7399) and the Wellcome Trust (grant WT096598MA). TJN acknowledges partial support from the Scottish University Life Science Alliance and the National Institutes of Health (Physical Sciences in Oncology Centers, U54 CA143682). The authors also acknowledge High Performance Computer resources partially supported by the Wellcome Trust (Centre Grant 083524).

## Additional information

### Funding

| Funder | Grant reference number | Author |
|---|---|---|
| Human Frontier Science Program | RGP-0038 | Luca Albergante |
| Cancer Research UK | C303/A7399 | J Julian Blow |
| Wellcome Trust | WT096598MA | J Julian Blow |
| Scottish University Life Science Alliance | | Timothy J Newman |
| National Institutes of Health | Physical Sciences in Oncology Centers, U54 CA143682 | Timothy J Newman |
| Wellcome Trust | 083524 | Luca Albergante, J Julian Blow, Timothy J Newman |

The funders had no role in study design, data collection and interpretation, or the decision to submit the work for publication.

### Author contributions

LA, Provided original concepts, Developed the data analysis tools, Performed the data analysis, Co-wrote the manuscript; JJB, TJN, Provided original concepts, Co-wrote the manuscript

## Additional files

### Supplementary files

• Supplementary file 1. Edges that result in the formation of at least one long feedback loop when inserted in the GM12878 GRN. The number of long feedback loops created and the length of the longest feedback loops are also indicated. Supplementary file 1 lists all the 48 edge insertions that result in the creation of long feedback loops when inserted in the GM12878 GRN. Note the high participation of JUNB and BATF as sources and of FOSL1/2 as targets. Three of these destabilizing interactions are actually observed in the K562 cancer cell line: from JUNB to BCLAF1, from BATF to EGR1, and from JUNB to EGR1. These interactions are highlighted in a bold character. The GRN of the GM12878 cell line contains 67 TFs and 59 interactions among them. Since in a network with 67 nodes there are 67*(67-1) = 4422 possible directed links with different source and targets, 4422-59 = 4362 additional regulations can be inserted between the TFs in the GM12878 GRN. Four regulatory connections between TFs are observed both in GM12878 and in K562: BCLAF1–> BHLHE40, CTCFL–> BHLHE40, EGR1–> JUNB, FOSL1–> JUNB. Of all the potential additional regulatory connections among the TFs of the GM12878 cell line, 56 are observed in the K562. Therefore, the likelihood of observing in K562 one of the 4362 potential regulations of GM12878 is 56/4362 ≈ 0.013. The likelihood of observing in K562 one of the potentially destabilising regulations of GM12878 is 3/48 = 0.062. Finally, two of the potentially destabilising interactions of the GM12878 have a particularly strong destabilization potential: an additional transcriptional regulation from JUNB to FOSL2 will create five long feedback loops and an additional transcriptional regulation from JUNB to E2F1 will create 3. Once again, our analysis highlights genes that may play a key role in cancer progression (*Kouzarides and Ziff, 1989*; *Engelmann and Pützer, 2012*).

• Source code 1. R Functions used to generate the random networks, to count the number of feedback loops, and to count the number of incomplete feedback loops.

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
