## [Decision Letter]

Thank you for sending your work entitled “Buffered Qualitative Stability explains the robustness and evolvability of transcriptional networks” for consideration at *eLife.* Your article has been favorably evaluated by Detlef Weigel (Senior editor) and 3 reviewers.

The editor and the other reviewers discussed their comments before we reached this decision, and the Senior editor has assembled the following comments to help you prepare a revised submission.

Specifically, a major condition for ultimately accepting the work would be to demonstrate that the observation of GRNs in the current setup containing very few long paths does not simply follow from the observation in the Luscombe et al. reference, that most regulatory interactions in response to a change in “exogenous” conditions will be fast and the underlying paths thus very short. This is also critical with respect to the influence of time delays. In this context, one needs to know how the size of the GRN in this work compares with that in other studies.

Another critical aspect of the work is the comparison to randomized networks. The naive randomization used has been shown to yield networks that are fundamentally different from real networks. Why did you not use a degree-preserving model? The Alon lab has already reported on the apparent lack of cycles in the GRN of E. coli, but found it to be non-significant when using degree-preserving randomization (Shen-Orr et al., Nature Genet. 31:64, 2002).

Similarly, while three stability rules are mentioned, it seems that only one is tested, and another one, that the sign matrix should be invertible, is not considered. In addition, it should be shown why this one property is sufficient for robustness. Along these lines, stability is a necessary, but not sufficient requirement for robustness. This should be discussed as well.

In general, the reviewers found the work in places difficult to follow; it is recommended to show at least one large GRN in E. coli in some detail, and explain how exactly you determine that there are at most three links in any loop.

Some aspects of cellular physiology certainly require destabilizing motifs (e.g. oscillations, switches). Have you by any chance found physiologically significant destabilizing motifs?

Finally, please correct the figures about numbers of IFLs (any IFL of size k should also be an IFL of size k-1), or clarify the definition of an IFL.

Comments in full:

*Reviewer #1*:

The authors analyze GRNs for stability using rules from the theory of qualitative stability, particularly verifying that they do not contain directed cycles of length >= 3 and derivatives of this property. They conclude that GRNs under normal conditions are designed for stability.

The authors mention three stability rules, however they test only one (the third) and seem to omit a fourth one (the sign matrix should be invertible – see e.g. the May 1973 ref). The authors should justify the omission and better argue for not testing the other two. Their arguments about further (post-transcription) forces beyond regulation are not convincing as they would also imply that testing for the third rule (no cycles with >= 3 vertices) ignores such post-transcriptional effects (in fact Shen Orr et al justify the lack of cycles in this way exactly in their Nature Genet 31:64, 2002 paper). Not less important would be to prove why the network could be robust (under the qualitative stability theory) if only this one property is satisfied.

The observation that GRNs contain very few long paths underlies many of the results in this paper. Could this property follow from the observation in the Luscombe et al. ref. that most regulatory interactions are involved in “exogenous” conditions in which the transcriptional response should be fast and the underlying paths are very short?

A critical aspect of the manuscript is the comparison to randomized networks. The authors use a naive randomization (ER) which has been shown to yield networks that are fundamentally different from real networks. Instead I recommend the authors use the more-or-less standard degree-preserving model as e.g. used in the publications of the Alon lab. Notably (see also above) the Alon lab has already reported on the lack of cycles in the GRN of E. Coli, but found it to be non-significant when using the degree-preserving randomization (Shen-Orr et al., Nature Genet. 31:64, 2002).

Regarding the networks themselves *–* no details are given about their sizes *–* could it be that the larger the network the more it deviates from the stability criterion just by mere chance (and regardless of biological considerations)?

There is a rich literature on network motifs and the results of this paper should be contrasted to it, clarifying what are the new observations here and how they relate to previous ones. For example, the authors claim to find no cycles in the yeast GRN. However, the Luscombe et al. paper also studied this network and reports on a cell-cycle related cycle (Figure 3) where the TFs of each phase regulate the subsequent one. Another example is the Shen-Orr paper mentioned above.

There seems to be a problem with counting IFLs in random networks as an IFL of size k is also an IFL of size 2...k-1, hence the bars should decrease in height as happens for real networks (Figure 2).

The authors should backup their claim about constitutively expressed TFs with gene expression data showing that this is the case.

*Reviewer #2*:

First, let me say that I want *eLife* to publish this paper ultimately. It is one of the few papers that make a serious physics-based approach to a fundamental biological problem, the stability of biological networks, I think *eLife*, if it is supposed to represent a new approach to biology, needs these kinds of papers. You should accept it.

That said, I also do appreciate that it is essential that physics-based approaches to biology problems have to reach out to biologists. Although I am an experimental physicist doing biological problems, I found this paper a tough read and I think the authors could make more of an effort to make this accessible to both biologists and physicists.

In the course of teaching a grad level course on biological physicist I deliberately went off the beaten path to look at areas that are not normally covered by biological physics textbooks but are actually very important to biology. The subject of this paper, gene regulatory networks and their fundamental stability, is something I dipped into, before I had seen this MS. I don't think the present MS. really does justice to the already quite extensive literature out there, very physics based, on GRN and this needs to be addressed at the outset. The authors might claim too much credit for what they have discovered. I found that the review in Nature Genetics by Albert-László Barabási and Zoltán N. Oltvai “Network Biology: Understanding the cell’s functional Organization” to be an excellent introduction and to have it characterized as “predominantly descriptive, rather than predictive” to be rather harsh. Buffered Qualitative Stability (BQS) may be an interesting subject (since it is qualitative and not quantitative really I could lump it into the “predominantly descriptive” category as well). I think more credit has to be given to the ideas of scale-free networks, hierarchical architecture and hub connectivity.

Next, the results here somewhat appear by magic from some huge (I guess huge) computer code which I don't think the authors really explain. Figure 1 is really makes quite remarkable claims, saying that real biological networks have far-far fewer feed-back loops than a randomly generated graph. But when I look at the biological networks that I am familiar with, such as the SOS network in E. coli, an organism which is one of the test cases (Figure 1), it is by no means obvious to me that there are just 3 at most links to that network: it is quite complicated. We looked at some other networks, and again we found many links to a loop. Large loops actually seem common to the naive eye.

So, it would be extraordinarily useful if the authors slowed down a bit and showed one large GRN in *E. coli* in some detail, and explain how exactly they determine that there are under 3 links to the loop! I am not sure that the SOS response qualifies as a GRN, but I am curious how this works out, and I think that giving a test example in some detail would help what seems to be a powerful idea which seems to be actually new works out in practice.

It would be great if BQS could be applied to cancer cells, as the authors’ claim. But they are frank to admit that the data isn't really there. But I don't think you need analysis of ALL the GRNs, I think that is hopeless. Certainly a critical stress-response like p53 must have a huge data set in various cells. Why not examine that and tell us if it actually is compromised for stability via BQS in cancer cells? And show us how the analysis is done, so we can try to repeat it?

In sum, a fascinating and potentially very important paper, but maybe a bit too boastful, and not very clear to the pedestrian, so that may really blunt it's impact.

*Reviewer #3*:

This is a very interesting manuscript which analyses the robustness and the evolvability of Genetic Regulatory Networks (GRN) in bacteria, yeast and human cells including a transformed one. The authors use the theory of qualitative stability analysis developed originally in economics. The theory summarized briefly in the Materials and Methods section of the manuscript provides necessary conditions for stability of a system of interacting components based on the sign of the elements in the 'community matrix'. Briefly, the system is stable against perturbations if it lacks positive, double-negative and long feedback loops. By comparative and statistical analysis of cellular and random networks, the authors show that GRN's seem to avoid the destabilizing network motifs. The only exception to this rule is the analysed transformed cell line. The paper is well written and I recommend it for publication but I would like to hear the comments of the authors for the following points:

1) The authors’ argument somehow implies that stability equals robustness. I think that stability is a necessary requirement for robustness but not sufficient. After perturbation the system might move to a completely different state even if it is stable. It would be useful to comment on this issue.

2) The theory of qualitative stability analysis is developed for autonomous systems. The cellular GRN has characteristic time-delays because of transcription and translation. It is true that in bacteria these two processes are overlapping but in eukaryotes the time-delay could be very significant. There is a short note about time-delay in the manuscript, but it would be useful to expand how it could influence these results.

3) Certain aspects of cellular physiology certainly require destabilizing motifs (e.g. oscillations, switches). I wonder whether the algorithm used by the authors has found any physiologically significant destabilizing motif

---

## [Author Response]

*Specifically, a major condition for ultimately accepting the work would be to demonstrate that the observation of GRNs in the current setup containing very few long paths does not simply follow from the observation in the Luscombe et al. reference, that most regulatory interactions in response to a change in “exogenous” conditions will be fast and the underlying paths thus very short. This is also critical with respect to the influence of time delays. In this context, one needs to know how the size of the GRN in this work compares with that in other studies*.

*Another critical aspect of the work is the comparison to randomized networks. The naive randomization used has been shown to yield networks that are fundamentally different from real networks. Why did you not use a degree-preserving model? The Alon lab has already reported on the apparent lack of cycles in the GRN of E. coli, but found it to be non-significant when using degree-preserving randomization (Shen-Orr et al., Nature Genet. 31:64, 2002)*.

*Similarly, while three stability rules are mentioned, it seems that only one is tested, and another one, that the sign matrix should be invertible, is not considered. In addition, it should be shown why this one property is sufficient for robustness. Along these lines, stability is a necessary, but not sufficient requirement for robustness. This should be discussed as well*.

*In general, the reviewers found the work in places difficult to follow; it is recommended to show at least one large GRN in E. coli in some detail, and explain how exactly you determine that there are at most three links in any loop*.

*Some aspects of cellular physiology certainly require destabilizing motifs (e.g*. *oscillations, switches). Have you by any chance found physiologically significant destabilizing motifs?*

*Finally, please correct the figures about numbers of IFLs (any IFL of size k should also be an IFL of size k-1), or clarify the definition of an IFL*.

The first point was to discuss whether the absence of loops feedback does not simply follow from the need for regulatory interactions to be fast and thus the underlying paths to be very short. Although we agree that this may be an additional selection pressure reducing the number of long loops, we think it is unlikely to be the primary explanation, for various reasons. First of all, certain transcriptional changes take place over very long time scales, indicating that a fast response time is not always necessary. Moreover, each transcription factor of a transcriptional cascade is potentially post-transcriptionally controlled (e.g. from signalling pathways), and therefore the signal need not proceed linearly from the top to the bottom of the cascade, but instead genes at the bottom of a cascade could be activated independently. Additionally, it should be remembered that BQS still allows a long pathway (for example A-B-C-D-E) to have a short link between the start and end (A-E in this example); when we enumerated incomplete feedback loops we considered all the possible transcriptional paths through the network, whilst a fast transcriptional propagation may only involve the shortest paths. Finally, we note that several key features of BQS that we observe in GRNs, including the prevalence of buffered stable 3- and 4-node motifs, the lack of transcriptional hubs with similar numbers of incoming and outgoing links, and the high prevalence of unregulated TFs are independent of signal pathway length. We added a paragraph to the Discussion (new paragraph 3) explaining this point. We also added a paragraph in the “Condition for Qualitative Stability of networks and its application to GRN” subsection of Materials and methods (new paragraph 9) discussing the possible effect of time delays.

With respect to the second point, i.e. to clarify why we decided not to use a degree-preserving model, we have included two new paragraphs to the end of the subsection “Effect of different constraints on the generation of random networks” in Materials and methods to better justify the null model used and to clarify the connections between BQS and previous works on degree-preserving models. Since a detailed analysis would be too extensive for an article focused on robustness, we decided to not to include a graphical representation of our results, but we can provide such a representation to the editors and reviewer if desired. As discussed therein, a degree-preserving model is strongly constrained by the original network considered and, in the case of GRNs, is likely to result in a network carrying the seed of robustness. To test this hypothesis, we assessed the effect of the degree preserving model commonly used in motif analysis (56) for the *E. coli* GRN. This model reshuffles the original network in such a way to completely preserve the original in-degree and out-degree *for each single node*. Our results (not shown) indicate that feedback loops are relatively uncommon in degree preserving random networks, regardless of the number of rounds of rewiring. However, incomplete feedback loops were more abundant and longer when compared to the real GRNs, even though to a lesser extent than purely random networks. Finally, the distributions of 3- and 4-node motifs were perturbed in such a way as to decrease the number of buffered motifs and to increase the number of unbuffered ones. However, even after 70,000 rounds of rewiring, the degree-preserving algorithm was unable to generate networks displaying an equal number of the 3- or 4-nodes motifs for the classes highlighted in the article, in striking difference from a purely random model. Taken together, these observations support the idea that GRNs carry a strong signature of BQS and that randomised models seeded by the real GRNs, and constrained to be very similar to GRNs will inherit, at least in part, many features of BQS. Our results also support the idea that purely random models may be the ideal null models to explore pervasive features of networks and that when the property under examination is strongly embedded into the system the use of highly constrained models may be disadvantageous. As we have demonstrated throughout our article, by comparison to random networks and by the analysis of the degrees of single TFs, robustness is strongly embedded in GRNs, since not only they are practically free from long feedback loops, but they are also configured in such a way that this property is preserved under quite extensive perturbation of their topology.

The third point was to clarify why the absence of long loops is the single fundamental property for robustness out of the 4 rules for Qualitative Stability. To address this we have expanded the explanation presented in the “Conditions for Stability for networks and its applicability to GRN” subsection of Materials and methods. The rule requiring invertibility of the sign matrix is not a relevant feature due to its implication of biochemical identity. The rule prohibiting autocatalytic positive self-regulation is not relevant to systems with finite resources such as GRNs, since saturating effects on transcription (for example due to limiting numbers of RNA polymerases) along with protein degradation means that autocatalytic TFs will reach a stable steady-state rather than increasing to arbitrarily high values. This argument of finite resources obviating positive self-regulation has also been applied to the study of Qualitative Stability by May in ecosystems (52). The rule requiring the absence of double positive and double negative feedback loops requires more attention. Under biological constraints relevant for GRNs, isolated (i.e. not connected with other feedback loops formed by two or more nodes) double negative feedback loops are unstable only to the extent that they form switches that can exist in one of two stable states. Double positive feedback loops are potentially capable of a more complex behaviour. However, when considered in isolation with negative self-regulation, they generally display a switch-like behaviour. Remarkably, in both GRNs for which sign information is available these conditions are verified: in *P. aeruginosa* the only double positive feedback loop is isolated and in *E. coli* all non-isolated two-gene feedback loops are part of the potentially chaotic motifs discussed above, and are therefore not relevant to this particular stability arguments. Therefore the only scenario in which condition 2) potentially threatens Qualitative Stability in GRNs is a ‘daisy chain’ of linked 2-node feedback loops; in the single case where such a daisy chain is observed (in *P. aeruginosa*) both 2-node feedback loops are of the positive/negative type and so are compatible with Qualitative Stability. Since this analysis further supports the concept of BQS in GRNs, we have added a more extensive discussion on 2-node motifs to the subsection “BQS highlights critical network modules” in Results. We also agree that stability is a necessary, but not sufficient requirement for robustness. We thank the reviewer and editors for pointing this out and have added a new paragraph in the Introduction to clarify this point (new paragraph 3).

The fourth point was to make the arguments easier to follow and to include one large GRN. We have amended the text to make our reasoning as transparent and simple as possible, which has also been helped by responding to the reviewers’ other comments. We have included a new Figure 1, which shows the full GRN of *E. coli*. We also added the new Figure 6—figure supplement 1 and the new Figure 7—figure supplement 1 which show all the transcriptional interactions between TFs in *E. coli* and the human cell lines, with the interaction forming long feedback loops highlighted in red. We also included the code of the R function used to determine the number of feedback loops found in a network in the [Supplementary-material SD1-data].

The fifth point was whether we have found physiologically significant motifs, which are connected with aspects of cellular physiology that require destabilization (e.g. oscillations, switches). Such motifs are observed and discussed throughout the manuscript. In our analysis of *E. coli* and *M. tuberculosis* we highlighted how unstable motifs with a peculiar structure are connected with stress responses (subsection “BQS highlights critical network modules” of Results) and discussed the possible implications of this discovery (subsection “Implication for evolvability and drug design” of Discussion). Moreover, in analyzing dendritic cells responding to a pathogen (subsection “BQS and transcriptional response” of Results), we see that a potentially stable network moves into a less stable configuration. Although we see the appearance of some interesting potentially unstable motifs in this response, the data set is too small to make any confident prediction about them. In the updated “Conditions for Qualitative Stability” subsection of Materials and methods we also detail more extensively the possible sources of switches (new paragraph 5 and 6) and highlight how post-transcriptional control may contribute to instability (new paragraph 8). It is our plan to further investigate the sources of instability in GRN, but this will require a more dynamic approach and will be the subject of future work.

The sixth point concerned the number of incomplete feedback loops and their definition. We want to clarify that the figures are correct. The distribution of incomplete feedback loops is tightly interlocked with the topology of the network under consideration and is monotonically decreasing only for specific network topologies. The methodology used to derive the number of incomplete feedback loops is detailed in the “Derivation of the networks and statistical analysis” subsection of the Materials and methods. However, we appreciate that such a concept leads to counter-intuitive results when applied to complex network topologies. Therefore, to better illustrate how the number of incomplete feedback loops depends on the topology of the network under consideration, we included a new Figure 3—figure supplement 5, which shows how three networks, labeled A, B and C, with the same number of nodes and edges, but with different topologies, have different distributions of incomplete feedback loops. In particular, the figure highlights how the number of incomplete feedback loops of length 3 can be smaller (A), equal (B), or larger (C) than the number of incomplete feedback loops of length 2. We also included an additional sentence in “Derivation of the network and statistical analysis” subsection of the Materials and Methods to highlight how the results on the distribution of incomplete feedback loops must be treated with particular care.

*Comments in full*:

Reviewer #1:

*The authors analyze GRNs for stability using rules from the theory of qualitative stability, particularly verifying that they do not contain directed cycles of length >= 3 and derivatives of this property. They conclude that GRNs under normal conditions are designed for stability*.

*The authors mention three stability rules, however they test only one (the third) and seem to omit a fourth one (the sign matrix should be invertible – see e.g. the May 1973 ref). The authors should justify the omission and better argue for not testing the other two*.

As illustrated above, in response to the reviewer’s question we have revised and strengthened our arguments in the “Conditions for Stability for networks and its applicability to GRN” subsection of Materials and methods.

*Their arguments about further (post-transcription) forces beyond regulation are not convincing as they would also imply that testing for the third rule (no cycles with >= 3 vertices) ignores such post-transcriptional effects (in fact Shen Orr et al justify the lack of cycles in this way exactly in their Nature Genet 31:64, 2002 paper). Not less important would be to prove why the network could be robust (under the qualitative stability theory) if only this one property is satisfied*.

As we mention, preliminary analysis of post-transcriptional networks suggests that long loops are under-represented but are not absent in the way they are with GRNs. We also subscribe to the idea suggested in Shen-Orr et al. about the separation of timescales for transcriptional and post-transcriptional networks and we suspect that this represents a fundamental difference between the way that transcriptional and post-transcriptional networks are regulated. Our analysis of mouse dendritic cells also suggests that post-transcriptional mechanisms are employed by the cells to introduce transient instabilities that would speed up the response of the cell. We would also like to point out that for similar reasons as described in Shen-Orr et al. the motif distributions highlighted would be preserved under moderate augmentation of the network by post-transcriptional modifications. Additionally, due to the nature of the randomization process used, the results presented in Shen-Orr et al. are compatible with a network configured to strongly limit the creation of long feedback loops even when as much as 25% of the network is perturbed. Additionally, while we agree that in the general case all the rules of Qualitative Stability must be considered, as discussed in the Materials and methods section, the absence of long feedback loops appears to be the most relevant for GRNs. In future, we would like to analyse the extent that

BQS applies to post-transcriptional networks, but this has to be the subject of another paper.

*The observation that GRNs contain very few long paths underlies many of the results in this paper*. *Could this property follow from the observation in the Luscombe et al. ref. that most regulatory interactions are involved in “exogenous” conditions in which the transcriptional response should be fast and the underlying paths are very short?*

This is an interesting alternative (or possibly complementary) explanation for some of the features predicted from our BQS model. As discussed above, we have added a new paragraph to the Discussion (new paragraph 3) to consider this point.

*A critical aspect of the manuscript is the comparison to randomized networks. The authors use a naive randomization (ER) which has been shown to yield networks that are fundamentally different from real networks. Instead I recommend the authors use the more-or-less standard degree-preserving model as e.g. used in the publications of the Alon lab. Notably (see also above) the Alon lab has already reported on the lack of cycles in the GRN of E. Coli, but found it to be non-significant when using the degree-preserving randomization (Shen-Orr et al., Nature Genet. 31:64, 2002)*.

We performed degree-preserving analysis as requested by the reviewer and report the results in the last paragraph of Materials and methods. Our results (not shown, but which we can provide to the reviewer if desired) have been described above. Additionally, we would like to stress the even though the lack of long feedback loops was observed previously, to our knowledge, this observation has never been connected to the robustness properties of the networks, and has not led to a uniform treatment of network topology.

*Regarding the networks themselves – no details are given about their sizes – could it be that the larger the network the more it deviates from the stability criterion just by mere chance (and regardless of biological considerations)*?

We have added a table to Materials and methods to provide the information requested. There is no evidence to suggest that larger networks show an increased instability; if anything it is the other way around as the larger networks are slightly less dense.

*There is a rich literature on network motifs and the results of this paper should be contrasted to it, clarifying what are the new observations here and how they relate to previous ones. For example, the authors claim to find no cycles in the yeast GRN. However, the Luscombe et al. paper also studied this network and reports on a cell-cycle related cycle (*Figure 3*) where the TFs of each phase regulate the subsequent one. Another example is the Shen-Orr paper mentioned above*.

The results of the analysis of the GRN derived by Luscombe et al. is presented in Figure 2—figure supplement 1, Figure 3—figure supplement 1, Figure 4—figure supplement 1, Figure 4—figure supplement 1, and Figure 5—figure supplement 2. As highlighted in the caption of Figure 2—figure supplement 1, Luscombe et al. generated their GRN by combining information derived from different methodologies and experimental conditions. While we fully appreciate the importance of Luscombe et al. in shedding light on the changes associated with different transcriptional responses, as shown by the Garber et al. data discussed in our paper, the GRN is reorganized in response to stimuli. Therefore, the GRN derived by Luscombe et al. is likely to be a superimposition of the GRNs active under different conditions. This suggests a perceivable level of false positives. Consistent with this idea, the loop structure observed in Luscombe et al. (Figure 2—figure supplement 1) is very similar to the loop structure of Harbison et al. with a p-val threshold of 3*10^-3 (Figure 2—figure supplement 3), and, as remarked by Lee et al. such a threshold is associated with a moderately high level of false positives. Additionally, while long feedback loops are present in the GRN derived by Luscombe et al. (Figure 2—figure supplement 1), the BQS-supported motif distribution is still observable (Figure 5—figure supplement 2), consistent with a noisy GRN following BQS (Figure 5—figure supplement 3). Moreover, such penetrance of long feedback loops is observed neither in the Lee et. al. dataset, from which the Luscombe et al. dataset has been derived, nor in Harbison et al., which is the most recent ChIP-Seq dataset derived. We have also added a new sentence to the Discussion to highlight how Shen-Orr et al. reports that no feedback loops are observed in the GRN studied, consistent with our observations.

*There seems to be a problem with counting IFLs in random networks as an IFL of size k is also an IFL of size 2...k-1, hence the bars should decrease in height as happens for real networks (*Figure 2*)*.

Due to the methodology used (sub-isomorphisms counting) the distribution of incomplete feedback loops is directly connected with the topological structure of the network and it is not always true that their number is monotonically decreasing. As illustrated above, to better clarify this point, we have added additional statements to the Results and Materials and Methods, and have included the new Figure 3—figure supplement 5, which illustrates how different topologies result in different incomplete feedback loop distributions.

*The authors should backup their claim about constitutively expressed TFs with gene expression data showing that this is the case*.

This was our textual error. We have amended the text to say there is a relatively high proportion of TFs ‘that are not regulated by other TFs’.

Reviewer #2:

*First, let me say that I want eLife to publish this paper ultimately*.

We thank the reviewer for his positive view of our approach.

*It is one of the few papers that make a serious physics-based approach to a fundamental biological problem, the stability of biological networks, I think eLife, if it is supposed to represent a new approach to biology, needs these kinds of papers. You should accept it*.

*That said, I also do appreciate that it is essential that physics-based approaches to biology problems have to reach out to biologists. Although I am an experimental physicist doing biological problems, I found this paper a tough read and I think the authors could make more of an effort to make this accessible to both biologists and physicists*.

We have amended the text in several places to improve comprehensibility.

*In the course of teaching a grad level course on biological physicist I deliberately went off the beaten path to look at areas that are not normally covered by biological physics textbooks but are actually very important to biology. The subject of this paper, gene regulatory networks and their fundamental stability, is something I dipped into, before I had seen this MS. I don't think the present MS. really does justice to the already quite extensive literature out there, very physics based, on GRN and this needs to be addressed at the outset. The authors might claim too much credit for what they have discovered. I found that the review in Nature Genetics by Albert-László Barabási and Zoltán N. Oltvai ``’Network Biology:Understanding the cell’s functional Organization’ to be an excellent introduction and to have it characterized as ``predominantly descriptive, rather than predictive' to be rather harsh. Buffered Qualitative Stability (BQS) may be an interesting subject (since it is qualitative and not quantitative really I could lump it into the ``predominantly descriptive' category as well). I think more credit has to be given to the ideas of scale-free networks, hierarchical architecture and hub connectivity*.

We accept that in our enthusiasm we were perhaps a little dismissive of previous analyses. We have modified the Introduction as suggested to give more credit to previous work, particularly in the first paragraph of the Introduction and the first paragraph of the Discussion. We had already cited the paper mentioned by the reviewer.

*Next, the results here somewhat appear by magic from some huge (I guess huge) computer code which I don't think the authors really explain.*
Figure 1
*is really makes quite remarkable claims, saying that real biological networks have far-far fewer feed-back loops than a randomly generated graph. But when I look at the biological networks that I am familiar with, such as the SOS network in E. coli, an organism which is one of the test cases (*Figure 1*), it is by no means obvious to me that there are just 3 at most links to that network: it is quite complicated. We looked at some other networks, and again we found many links to a loop. Large loops actually seem common to the naive eye*.

*So, it would be extraordinarily useful if the authors slowed down a bit and showed one large GRN in E. coli in some detail, and explain how exactly they determine that there are under 3 links to the loop! I am not sure that the SOS response qualifies as a GRN, but I am curious how this works out, and I think that giving a test example in some detail would help what seems to be a powerful idea which seems to be actually new works out in practice*.

To prevent readers feeling our data appears ‘by magic’ we have added as suggested a new Figure 1, showing the *E. coli* GRN with TFs coloured red and the arrows colour-coded according to the number of genes regulated by the source TF, a new Figure 6—figure supplement 1, which shows the interaction among TFs in *E. coli* and a new Figure 7—figure supplement 1 which contrasts the interactions among TFs in the GM12872 and K562 cell lines. Readers can trace links between TFs and confirm the absence of long feedback loops. As the reviewer concedes, loops such as those occurring in the SOS network involve post-transcriptional links rather than the purely transcriptional links that we analyse here. In addition, we provide a description of the algorithms used in Materials and methods and have included the R source code ([Supplementary-material SD2-data]).

*It would be great if BQS could be applied to cancer cells, as the authors’ claim. But they are frank to admit that the data isn't really there. But I don't think you need analysis of ALL the GRNs, I think that is hopeless. Certainly a critical stress-response like p53 must have a huge data set in various cells*. *Why not examine that and tell us if it actually is compromised for stability via BQS in cancer cells? And show us how the analysis is done, so we can try to repeat it?*

We don’t think that comprehensive data in the right form is really available.

While large amounts of gene expression data are available, the reconstruction of GRNs from such data is difficult and affected by over-prediction, which results in very noisy GRNs unsuitable for a direct analysis. While a motif analysis could potentially be viable, an extensive investigation of the effect of noise on the network properties supported by BQS is an essential prerequisite. We think that such an investigation and the analysis of GRNs derived by methodologies less reliable than ChIP-Seq would be an interesting follow-up work for a further publication.

Reviewer #3:

*1) The authors’ argument somehow implies that stability equals robustness. I think that stability is a necessary requirement for robustness but not sufficient. After perturbation the system might move to a completely different state even if it is stable. It would be useful to comment on this issue*.

We agree with the reviewer’s point that we dealt rather too briefly with the different forms that robustness can take. As illustrated above, we have added a paragraph to the Introduction (new paragraph 3) explaining that in this work we are focusing specifically on the stability component of robustness.

*2) The theory of qualitative stability analysis is developed for autonomous systems. The cellular GRN has characteristic time-delays because of transcription and translation. It is true that in bacteria these two processes are overlapping but in eukaryotes the time-delay could be very significant. There is a short note about time-delay in the manuscript, but it would be useful to expand how it could influence these results*.

We agree that time delays are probably an additional reason why GRNs have so few long loops. As illustrated above, we have added a new paragraph to the Discussion (Discussion, paragraph 3) and a new paragraph to Materials and Methods (First subsection, paragraph 5), dealing with different aspects of time delays and pathway lengths. Additional comments related to this aspect are discussed in response to Reviewer #1 point 3.

*3) Certain aspects of cellular physiology certainly require destabilizing motifs (e.g. oscillations, switches). I wonder whether the algorithm used by the authors has found any physiologically significant destabilizing motif*?

As illustrated above, we highlighted how unstable motifs are connected with stress responses, discussed how double positive and double negative 2-node feedback loops are likely sources of switches, and illustrated how in dendritic cells responding to a pathogen, we see that a potentially stable network moves into a less stable configuration. Although we see the appearance of some interesting potentially chaotic motifs in this response, this particular data set is too small to make any confident prediction of specific destabilizing motifs. The development of further statistical analyses focused on the role of potential sources of instability will be the will the subject of future work.